# Intermittent turbulence contributes to vertical dispersion of PM$_{2.5}$ in the North China Plain: cases from Tianjin

Wei Wei[1], Hongsheng Zhang[2], Bingui Wu[3], Yongxiang Huang[4], Xuhui Cai[5], Yu Song[5], Jianduo Li[1]

[1]State Key Laboratory of Severe Weather, Chinese Academy of Meteorological Sciences, Beijing 100081, P.R. China
[2]Laboratory for Climate and Ocean-Atmosphere Studies, Department of Atmospheric and Oceanic Sciences, School of Physics, Peking University, Beijing 100081, P.R. China
[3]Tianjin Municipal Meteorological Bureau, Tianjin 300074, P.R. China
[4]State Key Laboratory of Marine Environmental Science, Xiamen University, Xiamen 361005, P.R. China
[5]State Key Joint Laboratory of Environmental Simulation and Pollution Control, Department of Environmental Science, Peking University, Beijing 100081, P.R. China

*Correspondence to*: Hongsheng Zhang (hsdq@pku.edu.cn)

**Abstract.** Heavy particulate pollution events have frequently occurred in the North China Plain over the past decades. Due to high emissions and poor dispersion conditions, issues become increasingly serious during cold seasons. Although early studies have explored some potential reasons for air pollutions, there are few works focusing on the effects of intermittent turbulence. This paper draws upon two typical PM$_{2.5}$ (particulate matter with diameter less than 2.5 mm) pollution cases from the winter of 2016–2017. After several days of gradual accumulation, the concentration of PM$_{2.5}$ near the surface reached the maximum as a combined result of strong inversion layer, stagnant wind and high ambient humidity and then sharply decreased to a very low level within a few hours. In order to identify the strength of turbulent intermittency, an effective index, called Intermittency Factor (IF), was proposed by this work. The results show that the turbulence is very weak during the cumulative stage due to the suppression by strongly stratified layers; while for the stage of dispersion, the turbulence is highly intermittent and not locally generated. The vertical structure of turbulence and wind profiles confirm the generation and downward transport of intermittent turbulence associated with low-level jets. The intermittent turbulent fluxes contribute positively to the vertical transport of particulate matter and improve the air quality near the surface. This work brought up a possible mechanism of how intermittent turbulence affects the dispersion of particulate matter.

## 1 Introduction

In the winter of 2016–2017, severe air pollution events haunted the North China Plain, affecting more than 1/5 of the total population in China (Ren et al., 2017). Particulate pollution, especially PM$_{2.5}$ (particulate matter with diameter less than 2.5 mm) pollution, has become the foremost problem, considering its adverse impacts on human health (Dominici et al., 2014; Nel, 2005; Thompson et al., 2014; Zheng et al., 2015b).

Naturally, researchers are alarmed by these issues and want to understand the potential reasons. Some works (Wang et al., 2010; Zhang et al., 2016) reveal the effects of the increasing consumption of fossil fuel and the production of secondary pollutants. Meanwhile, it is reported that climate change (Yin et al., 2017; Yin and Wang, 2017) and synoptic circulation (Zhang et al., 2017; Miao et al., 2017; Ye et al., 2016; Zhang et al., 2012; Zheng et al., 2015a; Jiang et al., 2015) are of great importance in the transport of pollutants as well. Air pollution is essentially a phenomenon of the atmospheric boundary layer (ABL) and is strongly affected by the thermodynamic and dynamic structure of the ABL (Bressi et al., 2013; Gao et al., 2016; Tang et al., 2016). The spatial and temporal structures of turbulent motions have a dominant influence on the local air quality from the hourly scale to the diurnal scale (Shen et al., 2017). However, most of the works (Petäjä et al., 2016) focus on the feedback between aerosol, turbulent mixing and boundary layer, with little discussion on the dynamic effect of turbulence on the transport of particulate matter, not to mention the intermittent turbulence under strongly stable conditions. In fact, severe particulate pollutions tend to frequently occur in cold seasons in northern China (Sun et al., 2004; Zhang and Cao, 2015), during which the stratification of the ABL is more stable (Wang et al., 2017) and the turbulent mixing is relatively weak and intermittent in both temporal and spatial scales (Klipp and Mahrt, 2004; Mahrt, 2014). A series of works (Helgason and Pomeroy, 2012; Noone et al., 2013; Vindel and Yagüe, 2011) have confirmed that the intermittent turbulence accounts for a large amount of the vertical momentum, heat and mass exchange between the surface and the upper boundary layer, implying that intermittent turbulence may be one of the key factors in the pollutant dispersion.

The intermittency of velocity fluctuations comes in bursts (as shown in Fig. 2 in Frisch, 1980), which means that turbulent intermittency is non-stationary and has no specific time scale. Moreover, the turbulence in the ABL is inherently nonlinear (Holtslag, 2015) and has complex interaction with other motions, such as low-level jets, gravity waves, solitary waves and other non-turbulence motions (Banta et al., 2006; Sun et al., 2015; Terradellas et al., 2005). To date, different methods have been applied to describe the levels of intermittency, such as the flatness (Frisch, 1995), FI index (Flux Intermittency, Eq. (9) in Mahrt, 1998), wavelet analysis (Salmond, 2005) and so on. Given the non-linearity and non-stationarity of intermittent turbulence in the ABL, we conduct our study using a new technique, the so-called arbitrary-order Hilbert spectral analysis (arbitrary-order HSA, Huang et al., 2008), which has been successfully applied into the analyses of turbulence (Huang et al., 2009, 2011; Schmitt et al., 2009; Wei et al., 2016, 2017). It should be noticed that, the target of this work is not to compare the cons and pros of different methods but to study the turbulent intermittency in the ABL with the help of an effective method. The methodology is generalized and the advances are clarified in Sect. 2.2.

Based on these considerations, this work mainly aims at:

1) quantifying the turbulent intermittency in the ABL using the arbitrary-order HSA technique;

2) revealing a possible mechanism of the dispersion of near-surface $PM_{2.5}$ from a viewpoint of intermittent turbulence.

In the following text, the data and method are introduced in Sect. 2. Then Sect. 3 discusses our results in detail, including an overview of the cases, the behavior of turbulence intermittency and its contribution to the pollutant transport. The last Sect. 4 is a conclusion.

## 2 Data and Method

### 2.1 Observation

Tianjin (39.00° N, 117.21° E, altitude 3.4 m) is the largest coastal city in the North China Plain with a population of more than 1.5 million, covering an area of 11,300 km$^2$. Tianjin is located to the southeast of Beijing, the capital of China, and
neighbors Bohai Sea to the east (Fig. 1). Due to the rapid urbanization in the past decades, Tianjin has a typical urban underlying terrain.

Observations in this work include three parts: 1) a 255-m meteorological observation tower for the measurement of turbulence; 2) a CFL-03 wind-profile radar (WPR) for the boundary-layer wind field; and 3) a TEOM 1405-DF system for the monitor of particular matter. The 255-m meteorological observation tower is situated in the Tianjin Municipal
Meteorological Bureau, equipped with three levels (40, 120, and 200 m) of sonic anemometers (CSAT, CAMPBELL, Sci., USA) operating at a sampling frequency of 10 Hz. In addition, the observation (HMP45C, CAMPBELL, Sci., USA) at 15 levels is also used to analyze the behavior of relative humidity (RH) and temperature. The Tianjin Municipal Meteorological Bureau is located in a residential and traffic area and the buildings around the 255-m meteorological observation tower are typically 15-25 m in height (Ye et al., 2014). In order to avoid affecting the residential zone, the CFL-03 boundary-layer
WPR is mounted nearly 10 km away from the Tianjin Municipal Meteorological Bureau to the west. The 1405-DF TEOM system is located nearly 2.3 km away from the 255-m tower to the east and installed at a height of 3 m to monitor the surface PM$_{2.5}$. Detailed information is listed in Table 1.

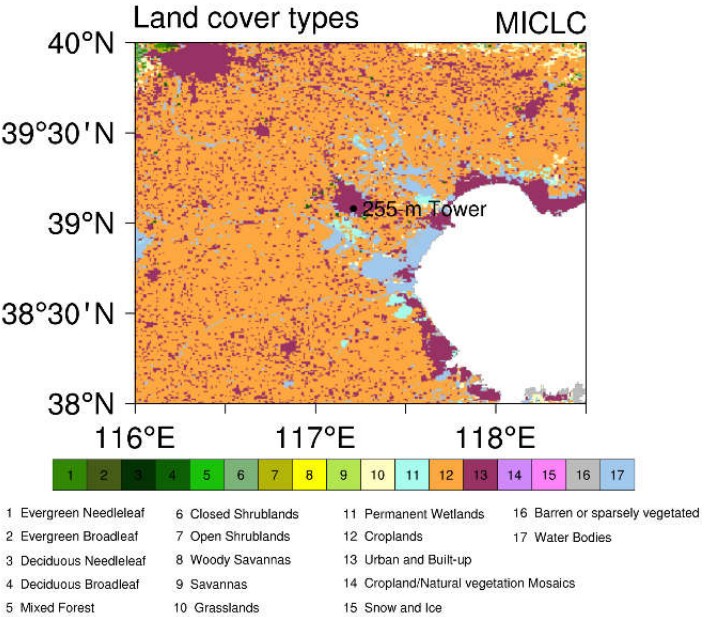

**Figure 1: Landuse map around the site. The black dot denotes the location of the 255-m meteorological observation**
**tower.**

**Table 1. Performance characteristics of instruments**

| Instrument | Height | Variables | Sampling resolution | Range | Accuracy |
|---|---|---|---|---|---|
| Sonic anemometer-thermometer, CSAT3[a] | 40, 120, 200 m | 3-D wind speed $(u_{x/y/z})$, Sonic virtual temperature | 0.1 s | $u_x$, $u_y$: $\pm 65.536$ m s$^{-1}$ $u_z$: $\pm 8.192$ m s$^{-1}$ $c^{[e]}$: 300–366 m s$^{-1}$ (-50–60°C) | $u_x$, $u_y$: $< \pm 4$ cm s$^{-1}$ $u_z$: $< \pm 2$ cm s$^{-1}$ |
| HMP45C[a] | 15 levels[b] | Temperature, Relative humidity | 15 s | -40–60°C 0–100% | $\pm 0.2$°C $\pm 2\%$ (<90%) $\pm 3\%$ (>90%) |
| 1405-DF TEOM[c] | 3 m | PM$_{2.5}$ | 1 h | 0–1,000,000 μg m$^{-3}$ | $\pm 7.5\%$ |
| CFL-03 boundary layer WPR[d] | < 5,000 m | Horizontal/vertical wind speed ($U_{h/v}$), wind direction | Temporal: 10 min Vertical: 100 m | $U_h$: 0–60 m s$^{-1}$ $U_v$: $\pm 20$ m s$^{-1}$ Direction: 0–360° | $U_{h/v}$: 0.1 m s$^{-1}$ Direction: $\leq 10$° |

[a] CAMPBELL, Sci., USA
[b] 15 levels: 5, 10, 20, 30, 40, 60, 80, 100, 120, 140, 160, 180, 200, 220, and 250 m
[c] Thermo Fisher Scientific, USA
[d] China Aerospace Science & Industry Corp
[e] c is the sound of speed

Turbulence observations from the 255-m tower were obtained at a sampling frequency of 10 Hz. Then quality control was applied to all the data (Zhang et al., 2001), such as error flag, spike detection, cross wind correction, spectral loss correction, sonic virtual temperature correction, density fluctuation correction, and coordinate rotation. If more than 20% points within a given 30-min time series were detected as outliers, then this 30-min observation was discarded. An averaging time length of 1 min was applied to calculate turbulent fluctuations and fluxes, given the small size eddies under stable conditions. The friction velocity $u_*$ reads in its form as $u_* = (\overline{u'w'}^2 + \overline{v'w'}^2)^{1/4}$, where u'/v'/w' represent the longitude, lateral and vertical fluctuation of wind vector. The turbulent kinetic energy (TKE) is given by TKE $= (\overline{u'^2} + \overline{v'^2} + \overline{w'^2})/2$ and the stability function uses z/L $= -\kappa z g \overline{w'\theta'}/\bar{\theta} u_*^3$, in which L $= -\bar{\theta} u_*^3/\kappa g \overline{w'\theta'}$ is Obukhov length, θ is potential temperature, z is observation height, g is gravitational acceleration and κ is von Karman constant with a value of 0.4 here. The data quality of CFL-03 boundary-layer WPR was checked to avoid the effects of poor quality of data. First, data below 200 m were removed due to the interference of surrounding environment, including trees and buildings. Then each vertical profile was checked through and points with larger than 2.5 standard deviations were regarded as outliers and discarded. A profile was discarded if more than 40% of the data points were outliers or missing (Wei et al., 2014).

Based on an overall consideration of data quality and severity of air quality, two cases happening in the winter of 2016–2017 were identified to study the relationship between intermittent turbulence and pollutant dispersion. The first one persisted for 5 days from 00:00 on 23 November 2016 to 00:00 on 28 November 2016, which is marked as Case-1 for convenience purposes. The second case, that is, Case-2, is from 00:00 on 23 January 2017 to 00:00 on 30 January 2017. All of the time in this work refers to Beijing Time.

## 2.2 Method

The flow in the ABL is highly nonlinear and non-stationary. In order to deal with the nonlinear and non-stationary time series, we adopted a relatively new technique called the arbitrary-order Hilbert spectral analysis (arbitrary-order HSA, Huang et al., 2008), which is based on the Hilbert-Huang transform (Huang et al., 1998, 1999). The primary reason why the arbitrary-order HSA is used in this work is that this method satisfies locality and adaptivity which are two necessary conditions for the study of nonlinear and non-stationary time series (Huang et al., 1998). Based on the arbitrary-order HSA, we proposed an index, called intermittency factor (IF), to quantify the level of turbulent intermittency, which is assumingly more effective compare with some classic quantities. To investigate the effects of vertical mixing in the dispersion of air pollutants, a set of vertical wind fluctuation at 10 Hz obtained by the sonic anemometers were drawn upon in this study. A brief introduction to the method is mathematically described in this part. For detailed information, one can refer to the work by Huang et al. (2008).

Firstly, a 30-min vertical wind-speed signal X(t) is separated into a group of intrinsic mode functions $C_i(t)$ and a residual $r_n(t)$ according to the so-called empirical mode decomposition. Here, each intrinsic mode functions $C_i(t)$ meets two constraints: (i) the difference between the number of local extrema and the number of zero-crossings must be zero or one, and (ii) the running mean values of upper and lower envelops are zero. The decomposition process is as follows (Huang et al., 1998, 1999):

1) The first step is to form the upper envelope $e_{max}(t)$ based on the local maxima of 30-min X(t) using the cubic spline interpolation. The lower envelope $e_{min}(t)$ can be constructed following the same method.
2) Then, one can define the mean $m_1(t) = (e_{max}(t) + e_{min}(t))/2$ and the first local signal $h_1(t) = X(t) - m_1(t)$.
3) So far, $h_1(t)$ is checked whether it meets the two constraints of intrinsic mode functions. If yes, $h_1(t)$ is the first intrinsic mode function $C_1(t) = h_1(t)$ and is taken away from X(t) to obtain the first residual $r_1(t) = X(t) - C_1(t)$. Then $r_1(t)$ is treated as the new signal to begin with step 1). If $h_1(t)$ does not meet the above constraints, the first step is repeated on $h_1(t)$ to define the lower and upper envelopes and further the new local detail until $h_{1k}(t)$ is the first intrinsic mode function $C_1(t) = h_{1k}(t)$.

Steps 1–3 are called 'sifting process'. To avoid over-sifting, the standard deviation criterion (Huang et al., 1998) is applied to stop this decomposition process. After n times of 'sifting process', one obtains a set of $C_i(t)$ and a monotonic residual $r_n(t)$.

At this point, the vertical wind fluctuation X(t) can be expressed as $X(t) = \sum_{i=1}^{n} C_i(t) + r_n(t)$. Then, each mode $C_i(t)$ is developed to obtain its corresponding analytical signal $C_i^A(t) = C_i(t) + j\tilde{C}_i(t) = A_i(t)\exp(j\theta_i(t))$ using Hilbert transform (Cohen, 1995), where the imaginary part reads as $\tilde{C}_i(t) = \frac{1}{\pi}\int_{-\infty}^{\infty}\frac{C_i(\tau)}{t-\tau}d\tau\frac{d\theta_i}{dt}$, and $A_i(t)$ and $\theta_i(t)$ are the instantaneous amplitude and phase. Also, one can define the instantaneous frequency as $\omega_i(t) = \frac{1}{2\pi}\frac{d\theta_i}{dt}$.

Note that the instantaneous amplitude $A_i(t)$ and frequency $\omega_i(t)$ are both a function of time, which means that a Hilbert spectrum $H(\omega, t)$ can be defined with $A_i(t)$ expressed in the space of frequency–time. So does the joint probability density function (p.d.f.) $p(\omega, A)$. If $H(\omega, t)$ is integrated with respect to time, one can get $H(\omega)$ which can be further expressed as $H(\omega) = \int p(\omega, A)A^2 dA$. If the power exponent of instantaneous amplitude is extended from 2 to q, one can define arbitrary-order Hilbert spectrum as $\mathcal{L}_q(\omega) = \int p(\omega, A)A^q dA$, where q $\geq$ 0 is the arbitrary moment.

In the case of scale invariance, the arbitrary-order Hilbert spectrum follows $\mathcal{L}_q(\omega) \sim \omega^{-\xi(q)}$ in the inertial subrange, in which $\omega$ is the frequency and $\xi(q)$ is the scaling exponent function. Under the assumption of fully developed turbulence, the distribution of scaling exponent function with the order $q$ is linear and meets $\xi(q) - 1 = q/3$, which is developed from $\xi(q) = \zeta(q) + 1$ (Huang et al., 2008, 2011), where $\zeta(q)$ is the scaling exponent function in q–order structure function $S_q(l) = \langle(\delta X(l))^q\rangle = \langle(X(l+l_0) - X(l_0))^q\rangle \sim l^{\zeta(q)}$, in which the angular bracket refers to spatial averaging and $l$ means distance. This exponent law is in agreement with Kolmogorov's hypothesis (K41 for short) and any intermittency would result in deviations from the theoretical $q/3$ (Basu et al., 2004). Based on this, we define an index IF as the deviation from the theoretical value at the maximal order: IF $= \xi(q_{max}) - 1 - q_{max}/3$. Due to the limited observation length, the maximal order $q_{max}$ is up to 4 in this study to avoid the difficulties and errors in the measurements of high-order moments (Frisch, 1995).

It is well acknowledged that the intermittent turbulence under stable conditions is characterized by sporadic bursts in a timescale of order $O(10)$ to $O(1000)$ sec. The statistically unsteady turbulence disobeys the assumptions of traditional theories (Poulos et al., 2002). For example, Fourier spectral analysis asks for a linear system and strictly stationary data; and the widely used wavelet transform is suitable for non-stationary signals but suffers when it comes to nonlinear cases (Huang et al., 1998). As one of the most important steps through this method, the empirical mode decomposition separates the

original time series into different modes based on its own physical characteristics without any predetermined basis, implying an intuitive, direct, adaptive, and data-based nature. And with the instantaneous information from the Hilbert transform, one can investigate the behavior of local events, which makes the Hilbert-based method more appropriate for the analyses of intermittent turbulence. This Hilbert-based scaling exponent function $\xi(q)$ has been applied into the analyses of turbulent intermittency in the ABL (Wei et al., 2016, 2017) and shown its effectiveness and validity.


## 3 Results and Discussion

### 3.1 Overview of Cases

Figure 2 illustrates the time series of different variables for two cases, including surface $PM_{2.5}$ concentration, wind vector, temperature, RH, horizontal wind speed, vertical wind speed, friction velocity $u_*$, TKE, and stability parameter z/L. From the distribution of $PM_{2.5}$, it can be seen that the concentration of $PM_{2.5}$ gradually increased to maxima (263 µg m$^{-3}$ for Case-1 and 412 µg m$^{-3}$ for Case-2) and then dropped to a low level within a few hours. Based on the concentration of $PM_{2.5}$, we can easily divide each case into two periods: one called the cumulative stage (CS) during which the particulate matter accumulates near the surface; the other named as the transport stage (TS) representing the stage when pollutants dissipate (Zhong et al., 2017). At this point, Case-1 can be separated into the CS from 00:00 on 23 November to 06:00 on 27 November 2016 and the TS from 06:00 on 27 November to 00:00 on 28 November 2016. Case-2 experienced two transitions from the CS to TS which happened at 00:00 on 26 January 2017 and at 00:00 on 29 January 2017, respectively. To distinguish these two transitions, the former is marked as Case-2A and the latter is Case-2B. Table 2 compares the values of mean and standard deviation of different variables between CSs and TSs. Generally, the mean concentration of $PM_{2.5}$ during the CS is much higher than that for the TS. For Case-1, wind at lower levels mainly comes from the south-east during the CS, while the dominant wind direction turns into west when it comes to the TS. Although the wind direction for Case-2 is seemingly unsteady in Fig. 2, the statistical the rose diagrams (see Fig. S7) confirm a similar result, with south-easterly flows dominating the CS and westers for the TS. This wind-direction pattern is in agreement with previous works (Zhang et al., 2017; Miao et al., 2017; Zheng et al., 2015a; Jiang et al., 2015). They found that south-easterly wind can bring the aerosols emitted by the surrounding cities to this region while the clean hours are normally characterized by strong high-pressure centers northwest of the polluted region in winter. However, in the region with densely distributed mega-cities (as in the case of Tianjin), because the upwind flows is polluted, mere advection may not be enough to disperse pollutants, thus resulting in persistent air pollution events (Zheng et al., 2015a; Chan and Yao, 2008).

In terms of temperature, it goes through gradual increase over CSs despite of its diurnal change. Furthermore, Fig. 3 depicts the distribution of Planetary Boundary Layer Height (PBLH) and the daily mean potential temperature profiles at 15 different heights, including change of θ over the CS (Fig. 3a–c) / TS (Fig. 3d–f) and the development during the whole polluted event (Fig. 3g). The Δθ at given height of CSs was calculated by subtracting the value of θ on the last day from that on the first day. And so it does for TSs. For Case-1, Δθ during the CS at the lowest level (5 m) is only 5.2 K. But for the top level at 250 m, Δθ is relatively larger with a value of 6.8 K. This result confirms that the warming of upper layers is stronger than that of lower layers, implying an increasingly stably stratified boundary layer during polluted days. Figs. 3b and 3c for Case-2 verify this conclusion as well. On the contrary, Δθ during TSs (Fig. 3d–f) presents a significant cooling at higher levels, denoting the collapse of inversion layer at the end of the polluted event. Taking Case-1 as an example, Fig. 3g depicts the evolution of inversion layer. It can be seen that the inversion layer was gradually enhanced from 23 to 27 November but quickly depressed on 28 November, which verifies the results of Fig. 3a–f. Fig. 3h illustrates the distribution of PBLH,

which is simulated with the Weather Research & Forecasting (WRF) Model (Zheng et al., 2015c). In Fig. 3h, the PBLH for Case-1 gradually decreased and reached its minimum on the night of 26–27 November. Then the PBLH redeveloped to higher than 1,300 m during the daytime of 28 November. Besides, an ambience with high relative humidity (RH) is favorable for the increase of $PM_{2.5}$ concentration in the ABL through secondary formation by heterogeneous reactions (Quan et al., 2015; Wang et al., 2012; Faust et al., 2017) and hygroscopic growth (Engelhart et al., 2011; Petters and Kreidenweis, 2008). For Case-1, the RH during the CS keeps high with a mean value of 53% but sharply falls into a very low level once entering the TS. Similar results can be found in Case-2.

During CSs, both horizontal wind and vertical wind are weak, implying unfavorable transport conditions. On the contrary, the strength of horizontal and vertical wind notably increases during TSs. These results are consistent with that of $u_*$, showing a total vertical momentum flux with a mean of 0.25 m s$^{-1}$ during the CS of Case-1 (0.19 and 0.29 m s$^{-1}$ for Case-2). While the values of $u_*$ are generally larger than 0.30 m s$^{-1}$ in the TS of both two cases. According to the classic TKE budget equation (Eq. 5.2.3 in Stull, 1988), the TKE is distinctively produced by the mechanical wind shear near the surface in the TS, resulting in strong turbulent mixing in the ABL, thus more effective transport of air pollutants. Another important term in the TKE budget equation is the buoyant production or consumption. The stability parameter z/L is used here to quantify the stratification of layers near the surface. Although the values of z/L are negative during the daytime, nocturnal z/L during the CS is notably larger than 1, which means that the consumption caused by buoyancy is dominant compared with the weak production by wind shear. The strongly stable stratification near the surface restrains the vertical turbulence mixing. The reduced mixing, together with emissions and production of secondary pollutants, result in a heavily polluted layer near the surface.

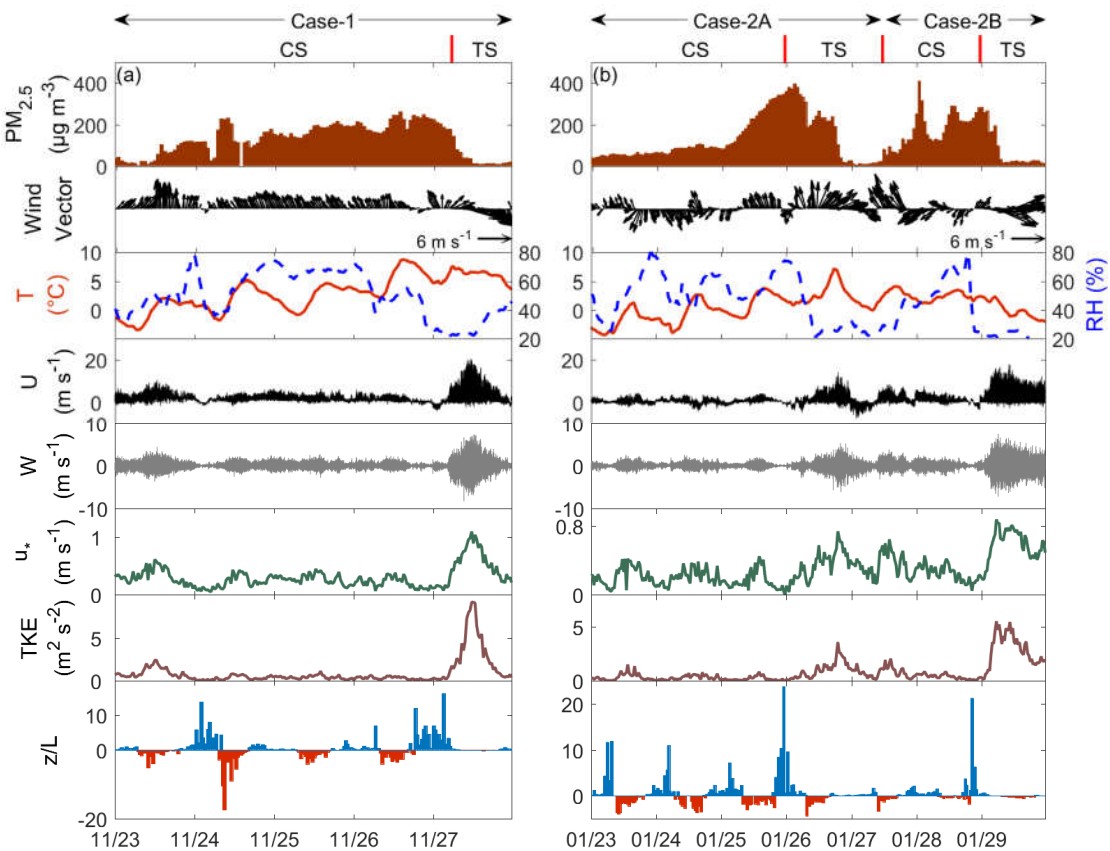

**Figure 2. Time series of surface PM$_{2.5}$, wind vector, temperature (T), relative humidity (RH), horizontal wind speed (U), vertical wind speed (W), friction velocity ($u_*$), turbulent kinetic energy (TKE), and stability parameter (z/L) for Case-1 in (a); for Case-2 in (b). The CS refers to the stage during which pollutants culminated and the TS represents clear days. All of the variables were observed at 40 m except for PM$_{2.5}$ concentration which is at the surface.**

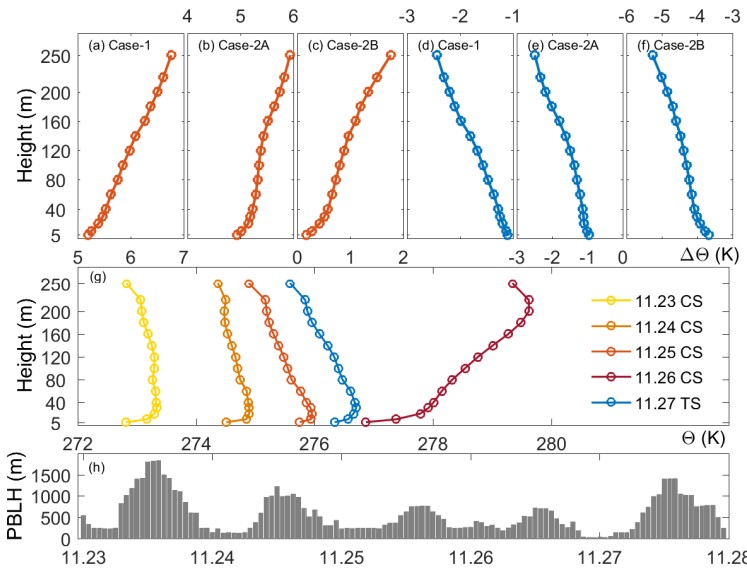

**Figure 3. Vertical distribution of daily mean potential temperature. The change of daily mean potential temperature of CSs is showed in (a)–(c) and (d)–(f) are for TSs. (g) illustrates the evolution of inversion layer of Case-1. (h) is the PBLH simulated with WRF Model.**

**Table 2. Mean and standard deviation (Mean ± SD) of key variables during different periods**

| Variables | CS | TS | CS | TS |
|---|---|---|---|---|
| | | Case-1 | | Case-2 |
| $PM_{2.5}$ ($\mu g\ m^{-3}$) | $145 \pm 71$ | $31 \pm 35$ | $139 \pm 101$ <br> $179 \pm 85$ | $104 \pm 138$ <br> $75 \pm 92$ |
| Temperature (K) | $275.6 \pm 2.9$ | $279.2 \pm 0.9$ | $272.7 \pm 2.3$ <br> $275.6 \pm 0.8$ | $275.4 \pm 1.8$ <br> $273.1 \pm 1.4$ |
| RH (%) | $53 \pm 15$ | $32 \pm 8$ | $54 \pm 12$ <br> $48 \pm 15$ | $35 \pm 17$ <br> $22 \pm 2$ |
| U (m s$^{-1}$) | $1.85 \pm 1.38$ | $3.78 \pm 2.90$ | $0.61 \pm 0.87$ <br> $0.67 \pm 2.03$ | $1.47 \pm 1.39$ <br> $3.23 \pm 1.98$ |
| Magnitude of W (m s$^{-1}$) | $0.28 \pm 0.26$ | $0.61 \pm 0.64$ | $0.23 \pm 0.22$ <br> $0.29 \pm 0.29$ | $0.36 \pm 0.35$ <br> $0.62 \pm 0.58$ |
| $u_*$ (m s$^{-1}$) | $0.25 \pm 0.12$ | $0.59 \pm 0.26$ | $0.19 \pm 0.10$ <br> $0.29 \pm 0.13$ | $0.33 \pm 0.15$ <br> $0.58 \pm 0.18$ |
| TKE (m$^2$ s$^{-2}$) | $0.50 \pm 0.43$ | $3.35 \pm 2.81$ | $0.28 \pm 0.93$ <br> $0.25 \pm 0.71$ | $0.54 \pm 0.42$ <br> $2.87 \pm 1.50$ |
| z/L at night | $1.70 \pm 2.74$ | $0.23 \pm 0.21$ | $2.18 \pm 3.74$ | $0.61 \pm 1.43$ |

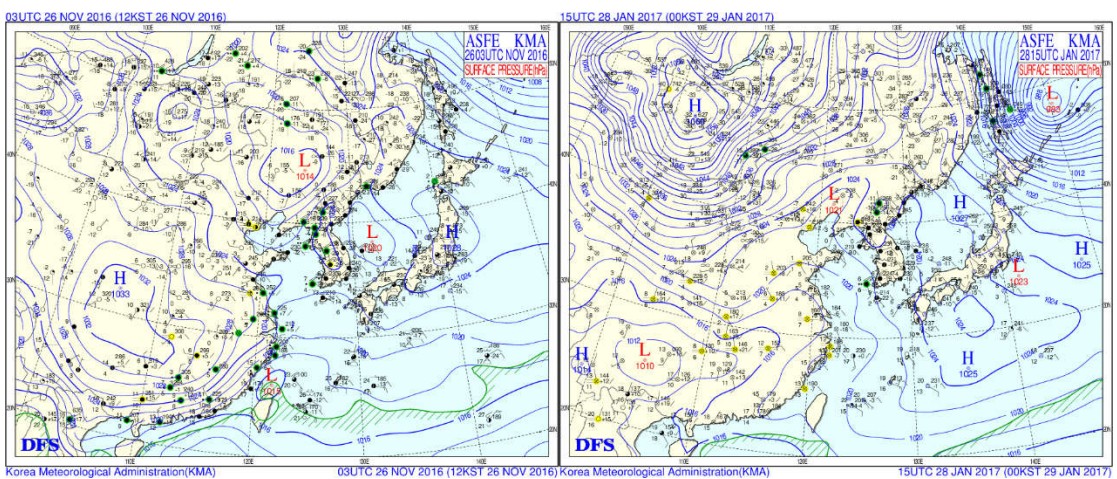

**Figure 4. Typical surface weather condition during CSs. Weather charts are from Korea Meteorological Administration.**

In addition to the meteorological parameters, synoptic weather conditions also play an important role in the formation and dissipation of heavy air pollutions (Zheng et al., 2015a). Fig. 4 takes two examples from Case-1 and Case-2 to illustrate the typical synoptic weather condition during CSs. In general, the accumulation of pollutants is accompanied by a low pressure system dominating the northern China. Under the control of cyclone system, this region is covered by sparse isobars and is controlled by stagnant wind field, resulting in unfavorable transport conditions.

## 3.2 Characteristics of intermittent turbulence

Considering the nonlinear and non-stationary nature of turbulent intermittency, it is imperative to choose an effective and reliable method before we implement the analyses. The arbitrary-order HSA used in this work to identify IF index meets the necessary conditions for the analyses of nonlinear and non-stationary time series, such as complete, orthogonal, local, and adaptive (Huang et al., 1998). Our previous works (Wei et al., 2016, 2017) have confirmed the validity of arbitrary-order HSA method in the identification of turbulent intermittency in the ABL.

As mentioned in Sect. 2.2, if turbulence in the ABL is fully developed, the Hilbert-based exponent scaling function $\xi(q)$ should follow the linear distribution of $\xi(q) - 1 = q/3$ (Huang et al., 2008). However, in the real world, there exit all kinds of instability mechanisms on different scales, such as the large-scale baroclinic instability and the small-scale convective instability (Frisch, 1980). Furthermore, under stable conditions, the very low boundary-layer height limit the development of eddies and the stagnant wind near the surface is not enough to maintain the turbulence mixing. Any of these mechanisms would destroy the statistical symmetries stored in the fully developed turbulence, resulting in deviations from K41's q/3 and

a set of concave curves in which the degree of the discrepancy of concave curves manifests the strength of turbulent intermittency. Figs. 5a–d present the behavior of $\xi(q) - 1$ at 40 m during different stages from two cases. Compared with the theoretical $q/3$, $\xi(q) - 1$ from CSs and TSs both shows deviations to some extent. However, the difference for TSs is much more obvious (in Figs. 5b and 5d), indicating stronger intermittency in the turbulence. Fig. 5e further gives a two-hour

example of vertical wind speed during 23:00 on 25 January to 01:00 and 26 January 2017, which covers the transition from the CS to TS in Case-2A. One noticeable feature is that the magnitude of vertical wind fluctuation significantly increases and is marked by strong burst lasting for nearly 25 min from 00:20 to 00:45 on 26 January 2017. On the contrary, the vertical wind speed is relatively weak and steady during the CS. But it should be kept in mind that the small deviations during CSs do not manifest fully developed turbulence but result from the very weak wind speed in the ABL, at which point the wind

shear is either absent or not strong enough to generate intermittency (Van de Wiel et al., 2003). The magnitude of vertical wind speed for CSs is generally less than 0.58 m s$^{-1}$. Under extremely stable conditions, the size of eddies may be too small to be detected by sonic anemometers (Mahrt, 2014).

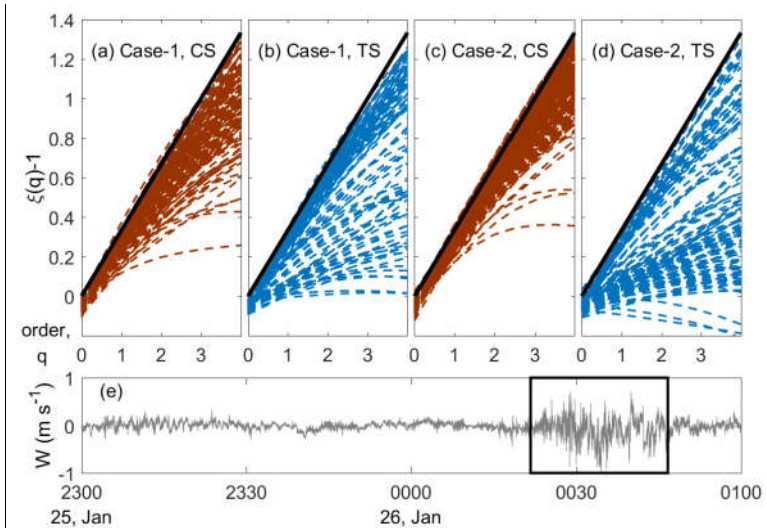

**Figure 5. Hilbert-based scaling exponent function at 40 m during different stages for (a)–(b) Case-1 and (c)–(d) Case-2, where each dashed curve represents the result of 30-min vertical wind speed signal and the black solid line denotes the K41 result q/3. (e) compares vertical wind fluctuation at 40 m between the CS (before 00:00 on 26 January 2017) and TS (after 00:00 on 26 January 2017). The latter shows apparent 'bursts' marked by the rectangular frame.**

In order to measure the strength of turbulent intermittency at different levels, an index named as IF was proposed by this work. Since IF is defined as the deviation from the theoretical $q/3$ at the maximal order $q_{max}$ (here $q_{max}$ equal to 4, see Sect. 2.2), the larger absolute values of IF indicate stronger turbulent intermittency at given height. Meanwhile, time with magnitude of vertical wind speed at 40 m less than 0.3 m s$^{-1}$ is marked to exclude stagnant wind. Fig. 6 illustrates the distribution of IF at three levels, compared with the concentration of PM$_{2.5}$ at the surface. Each sharp decline of surface

PM$_{2.5}$ concentration is accompanied by the abrupt change in values of IF. And IF at 40 m level is especially in agreement with the accumulation and dissipation of pollutants at the surface. The increasing deviations of IF when entering the TS manifest more intermittent turbulence, thus stronger vertical mixing in the ABL. In order to validate the results of IF, another two parameters to indicate the intermittency of turbulence were developed using the same data: one is kurtosis (Vindel et al., 2008) and the other is FI (Mahrt, 1998; Ha et al., 2007). The results of both kurtosis and FI are consistent with those of IF (see Fig. S8–9). In terms of the horizontal wind speed, the 40-m weak wind less than the threshold value (i.e. 5 m s$^{-1}$) proposed by Van de Wiel et al. (2012) implies that continuous turbulence is unlikely to occur near the surface (Fig. S10–11), which corresponds to the very stable turbulence regime-1 or regime-3 in Sun et al. (2012). Previous field results indicate that a significant proportion of vertical fluxes of heat, momentum, and mass under stable conditions come from intermittent bursts (Poulos et al., 2002). Fig. 7 further confirms the relationship between IF and $u_*$ or z/L, in which dots of strong turbulence ($u_*$) and weak stable stratification ($z/L \approx 0.1$) mainly come from the TS. Larger deviation of IF occurs accompanied by increasing turbulent strength when stability in the ABL becomes weaker. That is, intermittent turbulence (marked by large negative values of IF) leads to strong fluxes during the TS. The enhanced turbulent mixing caused by intermittent bursts contributes positively to the vertical transport of pollutants, improving the air quality near the surface. Besides, the points of intersection from the least-squares regression in Fig. 7 could denote the threshold beyond which the intermittency of turbulence arises under the mutual influence of dynamic and thermodynamic. The values of IF are -0.52 and -0.50 for Case-1 and Case-2, respectively. Hence, a cut-off value of IF (-0.50) can be identified to manifest the significant intermittency of turbulence. But it should be kept in mind that the absolute values of IF change from different heights and sites and this cut-off value of IF can only be used as a reference in the present study.

Besides, another feature of IF in Fig. 6 is that the occurrence of abruptly changed IF at different heights is not simultaneous. In general, the higher the level is, the earlier the intermittent turbulence happens and also the greater the deviations of IF are, which implies that the intermittent turbulence is generated at higher levels and then transported downward. We know that, under weakly stable conditions, turbulence in the ABL is continuous in both time and space, which is mainly dominated by wind shear at the surface. But for strongly stable cases, buoyancy prohibits the turbulent mixing, which enhances the surface radiative cooling, thus increasing the stratification of the ABL (Derbyshire, 1999). Such a positive feedback ultimately leads to the decoupling of boundary layer from the underlying surface, that is, the strong stratification prevents the turbulent exchange between the surface and the ABL. This decoupling could be ceased if there exists wind shear above the stable surface layer, at which point turbulence may be generated at upper levels and then transported downward to rebuild the coupling between the atmosphere and the surface (Mahrt, 1999). The decoupling is suddenly interrupted by the descending turbulence, resulting in intermittent bursts near the surface (Van de Wiel et al., 2012). It is reported (Smedman et al., 1995) that this downward transport of turbulence is related to the pressure transport term in the TKE equation, which means that Monin-Obukhov similarity theory is invalid for this case.

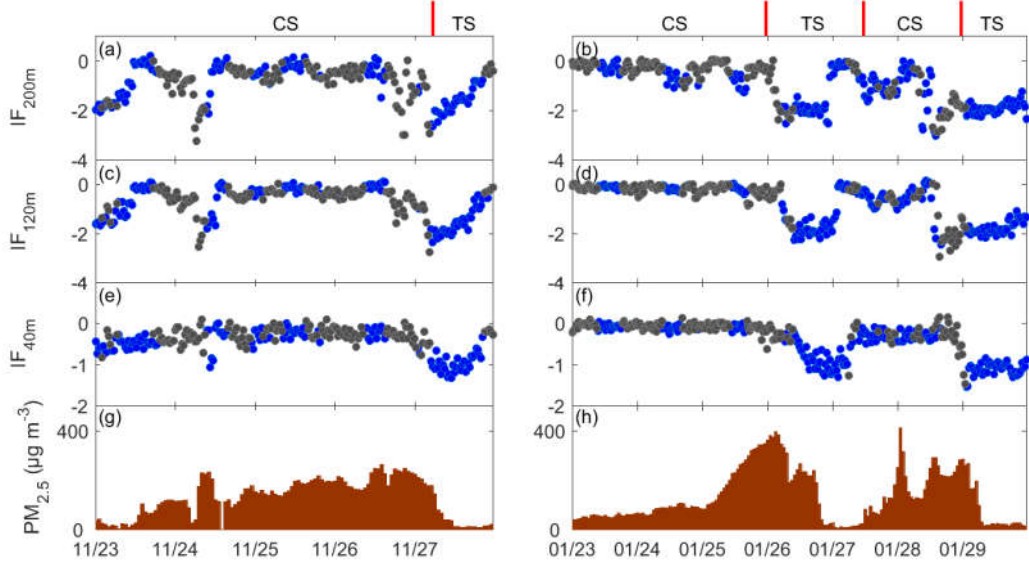

**Figure 6. Distribution of (a)–(f) IF at three levels and (g)–(h) concentration of PM2.5 at the surface. Left panel represents Case-1 and right panel for Case-2. Grey dots denote stagnant wind with vertical wind speed less than 0.3 m s⁻¹.**

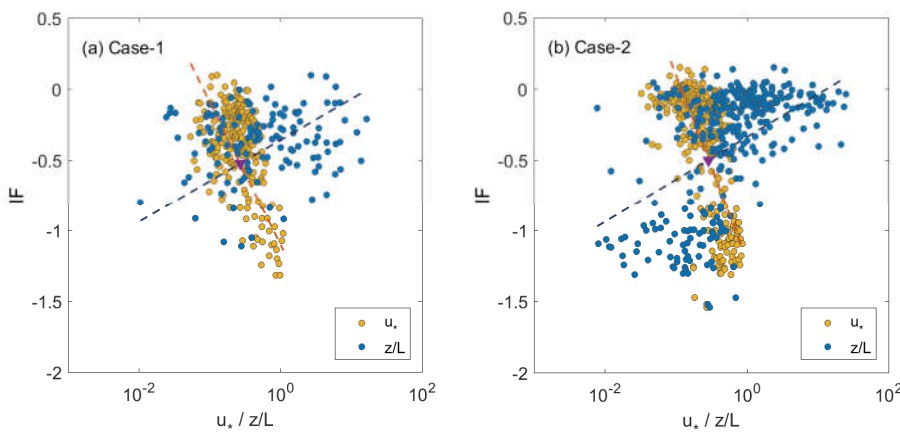

**Figure 7. Scatter plot of IF vs. $u_*$ and $z/L$ (night time) for (a) Case-1 and (b) Case-2 at 40 m. The dashed lines are the fittings from least-squares regression and the triangle marks the cross point.**

## 3.3 Mechanism and transport of intermittency

The reasons for intermittent turbulence in the ABL have not yet been well understood. Some potential causes include gravity waves (Sorbjan and Czerwinska, 2013; Strang and Fernado, 2001), solitary waves (Terradellas et al., 2005), horizontal meandering of the mean wind field (Anfossi et al., 2005), and low-level jets (LLJs, Marht, 2014; Banta et al., 2007; 2006; 2003).

5   2003).

According to the case overview in Sect. 3.1, it can be seen that the stratification of surface layers at night is fairly stable during CSs with values of z/L ≫ 1. Meanwhile, weak $u_*$ and TKE favor the accumulation of pollutants near the surface. In the case of decoupling, it is hard to generate turbulence through the interaction between the atmosphere and the surface. To detect the main sources of turbulence, Fig. 8 presents the height–time cross-section of horizontal wind speed under 2,000 m.

The lowest height range of WPR is at 200 m below which observations are seriously interfered by the hard-target returns of around buildings or trees. From Fig. 8a, strong wind occurs at night of 26 November 2016 with maximal wind speed larger than 15 m s⁻¹. For Case-2, there is also strong wind happening in the ABL right before the transitions between the CS and TS (Fig. 8b). After detecting the horizontal wind field, we found that the strong wind in the ABL is generally associated with the happening of LLJs. Fig. 9 takes three profiles as examples to illustrate the LLJ in the ABL. It is well-known that LLJs are an

important source of intermittent turbulence in the ABL, resulting in an 'upside-down' boundary layer structure (Mahrt, 1999; 2014; Poulos et al., 2002; Mahrt and Vickers, 2002; Banta et al., 2006; Balsley et al., 2003; Karipot et al., 2008). The vertical 'nose' shape of LLJs provides wind shear at upper levels, working as an elevated source of turbulent mixing. Then this turbulence is transported downward to the surface, resulting in non-stationary increase of turbulent mixing at lower levels. In this case, the vertical structure of 'upside-down' boundary layer is totally different from that of a traditional boundary layer.

For example, (a) the wind shear decreases with height first and then increases again due to the LLJ 'nose' at upper levels; (b) the strongest turbulence is not at the surface but aloft; (c) the transport of turbulence energy is downward. Fig. 10 presents the vertical turbulence structure across the tower layer for three dissipation nights, during which LLJs occurred. From Fig. 10a–b, it can be seen that the wind shear weakens in the layer between 40 and 120 m; then it increases when it comes to higher levels. In terms of the variance of vertical wind speed $\sigma_w^2$ (Fig. 10d–f), the maximal value of turbulence strength is

aloft rather than near the surface for all three nights, implying a turbulence source in mid-air. The vertical distribution of transport of turbulence energy further confirms the uplift of turbulence source. The values of the vertical transport of vertical velocity variance $\overline{w'^3}$ at three levels are negative generally, which means that the transport of turbulence energy across the tower layer is downward. It should be noticed that the magnitude of $\overline{w'^3}$ in Fig. 10h is not monotonously with height, implying a divergence layer of the downward transport of turbulence energy between 120 m and the top of the tower, which

suggests that this layer corresponds to the main source of the turbulence in the subjet layer (Mahrt and Vickers, 2002). In addition, the differences in phase and strength of intermittency at three levels in Fig. 6 also confirms that the wind shear associated with the LLJ 'nose' plays an important part in the generation and transport of turbulence in the ABL. Previous study (Wei et al., 2014) has revealed that the LLJ is a common phenomenon in Tianjin region, due to the combined effects of

plain terrain for inertial oscillation (Lundquist, 2003) and strong baroclinicity related to the land–sea temperature contrast offshore (Parish, 2000). In addition, the reduced convection in winter is helpful to maintain the LLJs even during the daytime. In a word, the frequent LLJs in Tianjin region are a key factor to understand the mechanisms of intermittent turbulence.

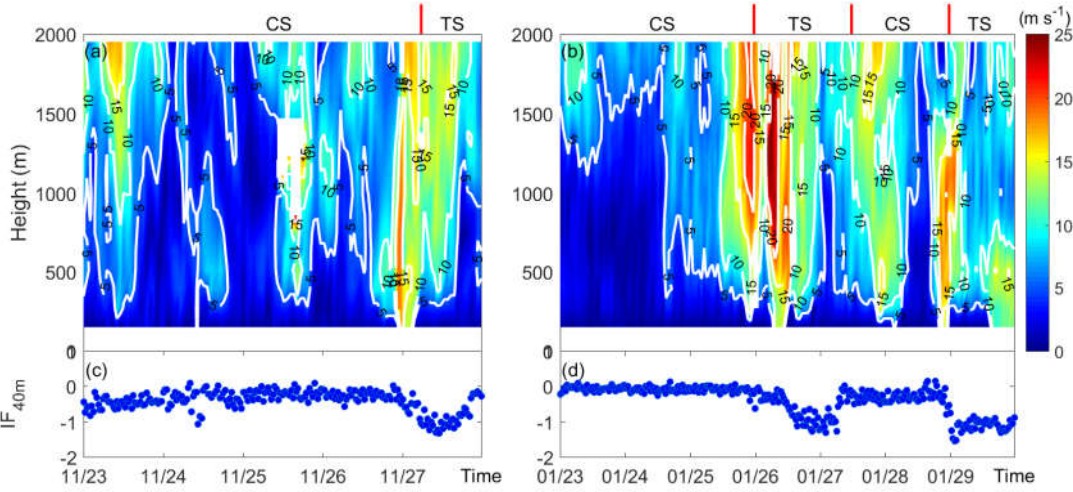

Figure 8. Height–time cross-section of horizontal wind speed observed by WPR for (a) Case-1 and (b) Case-2. (c)–(d) present the corresponding IF at 40 m.

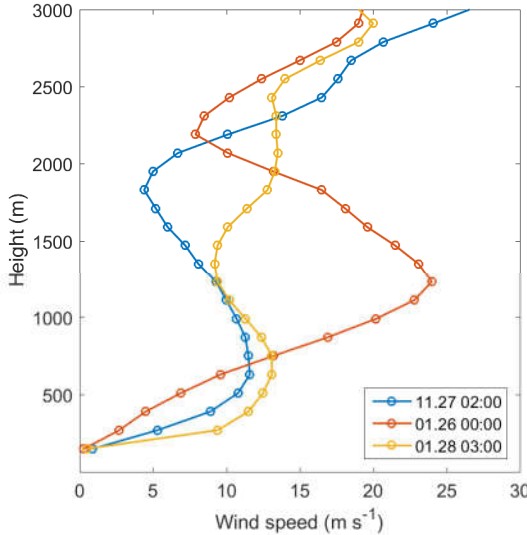

Figure 9. Sample LLJ profiles for three transitions between CSs and TSs.

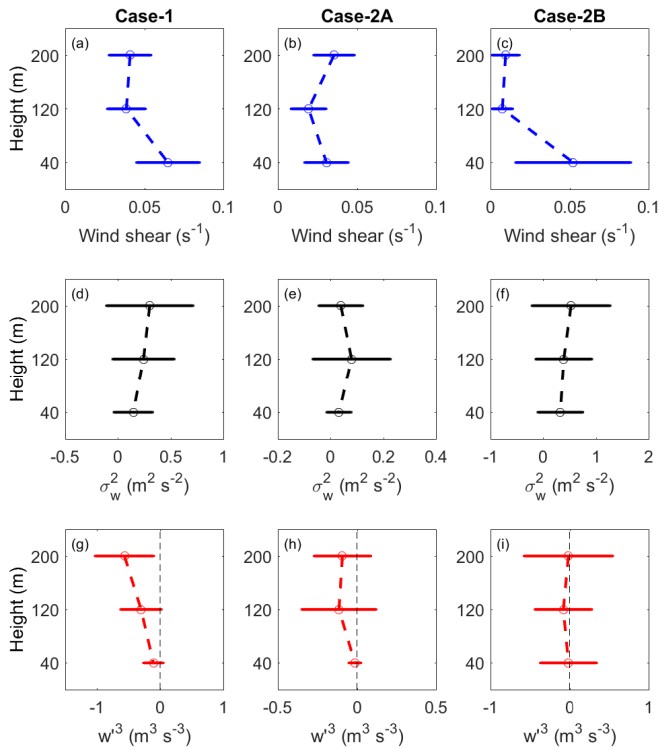

**Figure 10. Vertical structure of (a)–(c) Wind shear, (d)–(f) variance of vertical wind speed $\sigma_w^2$, (g)–(i) vertical transport of vertical velocity variance $\overline{w'^3}$ during (left) 00:00–00:06 LS on 27, Dec, 2016; (center) 00:00–00:06 LS on 26, Jan, 2017; (right) 00:00–00:06 LS on 29, Jan, 2017. The circle of errorbar denotes the mean value and the width of bar means the standard deviation.**

Finally, we summarize the mechanism of intermittent turbulence affecting the PM$_{2.5}$ concentration near the surface in a schematic figure in Fig. 11. In the beginning, the inversion layer near the surface enhances due to some favorable conditions including steady synoptic systems, stagnant wind, high temperature and high RH, which leads to the gradual accumulation of particles in the lower boundary layer. Such a process is named as the cumulative stage or CS, during which the turbulence

10   near the surface is too weak to transport pollutants upward. In this case, if there existed LLJs (or other motions) in the ABL, the turbulence could be generated by the strong wind shear associated with the LLJs and then transport downward, resulting in intermittent turbulence at lower levels. The suddenly increased vertical mixing is helpful for the abrupt dissipation of PM$_{2.5}$ near the surface.

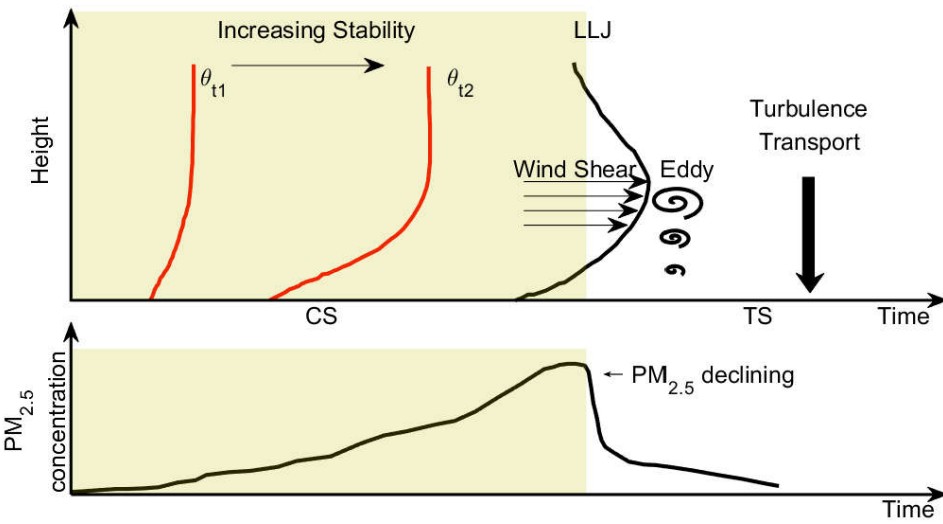

**Figure 11. Schematic of how intermittent turbulence affects the dissipation of PM$_{2.5}$ near the surface.**

## 4 Conclusion

In the winter of 2016–2017, severe air pollution events haunted the North China Plain. Extremely high concentration of PM$_{2.5}$ threatens the health of the large population living in this region. In this work, two cases in Tianjin (39.00° N, 117.43° E), China were drawn upon to study the effects of intermittent turbulence on the improvement of air quality near the surface. Observations from a 255-m tower, a CFL-03 WPR and a TEOM 1405-DF system were analyzed to investigate the features of boundary-layer structure and PM$_{2.5}$ concentration. In order to measure the levels of turbulent intermittency, an index,
called IF, was proposed based on the arbitrary-order HSA which is more suitable for the analyses of nonlinear and non-stationary signals.

At first, the inversion layer deepens with time. The stability function z/L keeps at high values (>> 1) at night and the stagnant wind field impedes the transport of pollutants. Additionally, high RH and steady cyclone system further aggravate the air pollution. All these factors result in the accumulation of pollutants near the surface.

On the other hand, the concentration of PM$_{2.5}$ undergoes rapid decrease during the dissipation periods, dropping from more than 250 μg m$^{-3}$ to less than 50 μg m$^{-3}$ within a few hours. The dispersion of pollutants coincides with the enhanced turbulent mixing in the ABL. The mean values of $u_*$ and TKE rise to 2–9 times those of the polluted periods. The Hilbert-based exponent scaling function $\xi(q)$ shows great deviations from K41's theoretical result of q/3 by a set of concave curves, indicating that the enhanced turbulence in the ABL when entering the TS is intermittent rather than continuous or fully
developed. Using the IF index derived from the vertical wind speed, the abrupt change in IF at 40 m is in agreement with the

sharp drop of PM$_{2.5}$ concentration. In addition, the occurrence and strength of intermittent turbulence differ with observation levels. In short, the higher the level is, the earlier the turbulence happens and the stronger the intermittency is, which implies that turbulence is mainly generated at upper levels and then transported to the surface. For 40 m, a cut-off value of IF (-0.50) indicates the initiation of strong turbulent intermittency in the ABL, while this is not a universal value and the threshold varies with different cases.

From the observation of WPR, LLJs were detected right before the dispersion of PM$_{2.5}$. Previous work (Marht, 2014) has pointed out that the stronger wind shear at the height of LLJ 'nose' can be a source of turbulence under the condition of decoupling between the atmosphere and the surface. In this work, LLJs play a key role in the generation of upper-level turbulence. The subsequent downward transport of turbulence leads to intermittent mixing near the surface, thus enhancing the vertical dispersion of PM$_{2.5}$ and improving the air quality.

*Data availability*. Data used in this study are available from the corresponding author upon request (hsdq@pku.edu.cn).

*Competing interests*. The authors declare that they have no conflict of interest.

*Acknowledgement*. This work was jointly funded by grant from National Key R&D Program of China (2016YFC0203300), the National Natural Science Foundation of China (91544216, 41705003, 41475007, 41675018). We also thank Dr. Gao Shanhong at Ocean University of China for providing historical weather-condition charts. Many thanks to Sr. Engr. Yao Qing and Liu Jinle at Tianjin Municipal Meteorological Bureau for their help with the data observation.

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
