# Peer review of "Intermittent turbulence contributes to vertical dispersion of PM2.5 in the North China Plain: cases from Tianjin"

_Atmospheric Chemistry and Physics, 2018_

## Referee Comment (RC1) · Anonymous Referee #1 · 22 Apr 2018

General comments:

In this paper an index, called intermittency factor (IF), is proposed to quantify the strength of intermittent turbulence in the atmospheric boundary layer (ABL) and applied to study the contribution of intermittent turbulence to vertical diffusion of PM2.5 in the North China Plain (NCP). The observational data are all from Tianjin, one of the megacities in the NCP. These data include high-resolution (10 Hz) 3D wind and virtual temperature data collected at 3 levels (40m, 120m and 200m) as well as temperature and RH at 15 levels on a meteorological tower, hourly measurements of PM2.5 at 3m, and horizontal and vertical wind data from a wind profiler radar. The strength of turbulence in the ABL and vertical profiles of wind and temperature are obtained and used in the analysis of variations in the PM2.5 concentration. The authors have not made
systematic investigations instead they have focused on two cases containing pollution cumulative stage (CS) and transport stage (TS). A new technique, the so-called arbitrary-order Hilbert spectral analysis (HSA) is used for the derivation of IF. The validity of arbitrary-order HSA method in the identification of turbulent intermittency in the ABL was confirmed by the authors in their previous studies. It is shown that intermittent turbulence generated by low-level jets (LLJs) and downwards transported is a key driver for the diffusion of PM2.5 near the ground.

The IF index proposed in this paper seems to be a good scale to describe turbulence intermittency and can be useful in the analysis of accumulation and dispersion of pollutants in the ABL. The results of this paper are generally sound and can better our understanding of the role of thermodynamic processes in the variations of PM in the ABL. This paper is within the scope of ACP and generally well written. I recommend publication of this paper in ACP after addressing some issues.

Major comments:

(1) There are substantial differences between the CS and TS in TKE, u*, and even W and U (Fig. 2). One can use these quantities in the explanation of accumulation and diffusion of PM2.5. Why do we need an IF index? In other words, what is the advantage or superiority of using IF compared with other quantities? This point should be discussed in the paper.

(2) Are there any significant correlations between IF and TKE as well as other parameters? If there are, they should be presented and discussed.

(3) Your measurements are from Tianjin, which is just west of the Bohai Bay. The emission and formation of PM2.5 over the land areas are much stronger than over the sea. Therefore, I guess air from the Bohai Bay was much cleaner. During each TS the prevailing wind was either southeasterly or northeasterly, different from that during the CSs. Did the change in horizontal air flow contribute also to the decrease in the PM2.5 concentration? And how significant?

(4) You show in Fig. 3 vertical profiles of changes in potential temperature during the CSs. I think similar results for the TSs should be presented and discussed as well. You may show how rapidly the stable condition formed during the CSs was broken by intermittent turbulence. In addition, some discussions about the evolution of the PBL height may be also good for a more complete picture.

(5) Your results and conclusions are based on cases study. I think it is better to add "cases from Tianjin" or similar subtitle. And "vertical diffusion" in the title can be questionable if you cannot prove that the decrease in PM2.5 was solely due to the vertical diffusion.

Minor points:

Page 1 line 21: What do you mean by "wind filed"? Wind profile?

Page 2 line 21: Define "FI".

Page 2 line 32: Delete "respectively".

Page 3 line 4: Change "east" to "southeast".

Page 3 line 11: I think "HMP45C" is the type name of the probe and should be put in the brackets.

Page 3 line 16: Was the TEOM system installed near the tower or the WPR? Please make it clear.

Table 1: "c: 300-366 m s-1". Is this the range of wind speed that the sonic anemometer can measure? 300 is a very strange number here.

Page 4 line 16: Change "poor data" to "poor quality of data".

Page 4 lines 19-21: What are the criteria for data that are suitable for this study?

Page 5 line 2: "local standard time" or "Beijing Time"?

Page 5 line 9: "On this basis"? It is not clear what is denoted.

Page 5 lines 11-12: Delete "(CSAT3, CAMPBELL Inc., USA)" because the same information is given on page 3.

Page 5 line 19: Do you mean the local maxima that are found within every 30-min periods?

Page 5 line 9, page 6 lines 12-13, and page 11 lines 18-19: You are proposing or defining IF at these three places. This is redundant. I think you should define the IF index at a suitable place and use it elsewhere.

Page 6 line 31 and page 7 line 1: "...increased to a maximum of 412 ug m-3 for PM2.5 and then dropped to a low level within a few hours no matter for Case-1 and Case-2". Please check you expression. I do believe the maximum values in Fig. 2a and Fig. 2b are all 412.

Page 9 Fig. 2: Please add some ticks on the Y-axes of Figs. 2a and 2b.

Page 11 line 2: CSs or TSs?

Page 11 Fig. 5: Does each concave curve represent a 30-min result? Please make it clear.

Page 11 line 24: Delete "site".

Page 13 line 2: "under stable conditions"? Are you not talking about the TS?
* * *

---

## Referee Comment (RC2) · Anonymous Referee #3 · 23 Apr 2018

Normally, intermittent turbulence is associated with the stable boundary layer (SBL). That is, turbulence strength varies when the background stratification is generally stable. When the stratification is totally wiped out by strong turbulent mixing for a relatively long period, the relatively strong turbulent mixing is not considered as part of a time series of intermittent turbulent mixing anymore. In this study, a period of strong turbulent mixing occurred at the end of all the three cases and each of them lasted for nearly a day. If the authors think the strong turbulent mixing period in each case is part of the time series of intermittent turbulence, this is definitely not what intermittent turbulence in the traditional definition.

Physically, intermittent turbulent mixing in the SBL can be generated by intermittent strengthening wind shear, which can be associated with low level jets (LLJs) as pointed

by the authors in the paper. However, this mechanism is not new; Banta et al. (2003, JAS; 2006, QJ; 2007 JAS) have investigated relationships between LLJs and turbulent mixing extensively.

In addition, from the title of the paper, it seems that the authors would address the role of intermittent turbulent mixing to the vertical dispersion (not diffusion, diffusion is for molecular movements) of PM2.5. However, the observation indicates that the intermittent turbulent mixing during the high PM2.5 period is not strong enough to disperse PM2.5 and the significant reduction of PM2.5 is observed at the end of each event when strong mixing arrives. Then what is the significance of intermittent turbulence during the stable period? What is the significance of the new intermittent turbulence index introduced here in comparison with simple parameters such as wind speed if wind shear is the key physical process for dispersing PM2.5?

Furthermore, because the stable boundary layer is known to be associated with weak winds when wind direction variations can be significant, how does wind advection contribute to the temporal variation of PM2.5 besides vertical dispersion of PM2.5 by turbulent mixing? Where is the high PM2.5 source?

---

## Referee Comment (RC3) · Anonymous Referee #2 · 27 Apr 2018

This manuscript investigates the role of intermittent turbulence in alleviating heavy pollution episodes that frequently occur in China. The papers includes a theoretical background and analysis of measurement data related to 2 pollution episodes. While the vast majority of the prior research on air pollution episodes in China has concentrated on the factors favoring the accumulation of pollutants, this paper investigates a phenomenon that helps to get rid of high pollution levels. As such, I think that this paper is original enough to warrant publication in ACP. I have a few issues that should, however, be addressed before the publication.

The authors introduce an Intermittent Factor (IF) which they use for explaining the effects of intermittent turbulence on the observations. I have a few comments on this. First, it seems that q is the key variable when determing IF. Therefore, the authors

should explain more explicitly what is the exact meaning of q, not just to mention that it is the power exponentof the instantaneous amplitude of something. Second, IF is defined such that it is zero for fully developed turbulence and negative if not. However, the exact value of IF does not tell anything for the reader. Would it be possible to provide some idea how to interpret the value of IF. How small (or large in absolute sence) should IF be for the intermittent turbulence to be important etc?

The discussion of Figure 5 in the beginning of page 11 is a bit confusing. The authors state that the difference for CSs is much more obvious. I do not understand this statement. By looking at the figure, the differ curves for TS show more spread than the curves for CS. So what are the authors referring to when discussion about differences? Also, figures 5a-d have the straight line for fully developed turbulence (faint solid black lines). This line should show up more clearly in the figure and it should be said that it is a solid line.

Please check out that all the used mathematical symbols are explained in the text.

A few grammatical issues:

Page 1, line 25 should read "particulate matter" Page 14, line 7: . . .we summarize. . .

———————————————

---

## Author Comment (AC1) · 4 Jul 2018

Dear reviewer, We all appreciate your hard work on this paper. These constructive opinions help to improve our work to a great extent. We did our best to respond to each comment and make this work well-organized. With the help of your detailed comments, some mistakes in the original manuscript were found and revised. Details are listed as follows:

Major comments: There are substantial differences between the CS and TS in TKE, u*, and even W and U (Fig. 2). One can use these quantities in the explanation of accumulation and diffusion of PM2.5. Why do we need an IF index? In other words, what is the advantage or superiority of using IF compared with other quantities? This

[Figure]

point should be discussed in the paper.

Response: Thank you for your comments. In previous works, some traditional variables (i.e. TKE, u*, and W/U) are commonly applied to indicate the behavior of turbulence. And the results in our work (Fig. 2) also show the relationship between those variables and the accumulation and diffusion of PM2.5 to some extent. Indeed, those quantities are useful for the description of turbulent characteristics including strength and variation, but fail to reveal the intermittency of turbulence. As mentioned in the introduction, the reason why we focus mainly on the influence of intermittent turbulence is that the intermittent turbulence accounts for a large amount of vertical fluxes in stable boundary layers but the discussion of the effects of intermittent turbulence on the transport of PM2.5 is still limited. At this point, we need an effective way to describe the characteristics of turbulent intermittency. The IF index was developed from the arbitrary-order Hilbert spectral analysis (arbitrary-order HSA, Huang et al., 2008). Compared with some classic methods (such as Fourier analysis and wavelet transform), the arbitrary-order HSA technique is intuitive, direct, and adaptive, with a posteriori-defined basis, from the decomposition method, based on and derived from the data, it is more appropriate for the analysis of nonlinear and non-stationary turbulence signals. Our some previous work (Wei et al. 2017) has addressed the intermittency of turbulence in the SBL using the arbitrary-order HSA technique. But we should admit that, as a newly-developed approach, arbitrary-order HSA still suffers some disadvantages. For example, the spline fitting and the end effects need more improvements. In the case of weak signals imbedded in stronger ones, differentiation should be applied if needed. In spite of these problems, HSA is still the best available nonlinear and non-stationary data analysis method so far. Based these considerations, we used the arbitrary-order HSA in this work to study the behavior of turbulent intermittency. The advantages of IF or arbitrary-order HSA are addressed in the revision according to your advice: "Based on the arbitrary-order HSA, we proposed an index, called intermittency factor (IF), to quantify the level of turbulent intermittency, which is assumingly more effective compare with some classic quantities." (page 5 lines 10-13) and "As one of the most important

steps through this method, the empirical mode decomposition separates the original time series into different modes based on its own physical characteristics without any predetermined basis, implying an intuitive, direct, adaptive, and data-based nature" (page 6 lines 22-24). Some more detailed information is given in the supplement: "The reason why the arbitrary-order HSA is applied to this work is that this method is more suitable for the analyses of nonlinear and non-stationary turbulent signals, compared with traditional techniques, such as Fourier analysis and Wavelet transform. It is known that, the data, whether from physical measurements or numerical modeling, most likely will have some problems: (a) the total data span is too short; (b) the data are non-stationary; and (c) the data represent nonlinear processes. While the Fourier analysis has some crucial restrictions: the system should be linear; and the data should be strictly periodic or stationary; otherwise, the resulting spectrum will make little physical sense. However, for lack of alternatives, Fourier spectral analysis is still applied to many kinds of data which may result in misleading results. On the other hand, the wavelet approach is essentially an adjustable window Fourier spectral analysis, with the basic wavelet function that satisfies certain very general conditions. As the traditional technique for the analysis of intermittency, the structure function is essentially associated with the Fourier decomposition, which means that the scaling exponent function $\zeta(q)$ has some limitations in the application of nonlinear and non-stationary turbulence signals." (page 4 lines 9-21 in the supplement) "As discussed by Huang et al. (1998, 1999), the arbitrary-order HSA technique is intuitive, direct, and adaptive, with a posteriori-defined basis, from the decomposition method, based on and derived from the data, it is more appropriate for the analysis of nonlinear and non-stationary turbulence signals. Since its introduction, the HSA method has been successfully applied into different fields, including climatology (Molla et al. 2006; Hu et al. 2014), meteorology (Karipot et al. 2009; Vincent et al. 2011) and oceanography (Chen et al. 2014), to name just a few. One of the authors (Wei et al. 2017) used arbitrary-order HSA technique to separate fine-scale and large-scale motions in the stable boundary layer (SBL) and obtained a better approximation to the Monin-Obukhov similarity the-

ory than using bandpass filtering method. Bases on these considerations and previous work, we believe that the arbitrary-order HSA technique is a suitable method for the study of turbulence intermittency in the SBL." (page 4 line 21 and page 5 lines 1-9 in the supplement)

(2) Are there any significant correlations between IF and TKE as well as other parameters? If there are, they should be presented and discussed.

Response: Thank you for your suggestion. We examined the relationship between IF and other quantities, including u* as the dynamic parameter and z/L as the thermodynamic variable, and the results are shown as follows:

Figure 1 Scatter plot comparing IF and other variables (u_* and nighttime z/L) for two cases at 40 m. The dashed lines are the fittings from least-squares regression and the triangle marks the cross point.

Generally, the values of IF decrease with increasingly stronger turbulence, which meets our expectation. Under extremely strong stable conditions, the turbulence in the ABL is suppressed, accompanied by very small dynamic quantities (such as u* in this case). At this point, the values of IF are nearly zero, representing the extremely weak turbulent fluctuation. With the increase of turbulence strength, the abstract value of IF rises, indicating that the relatively stronger turbulence in the ABL is intermittent but not continuous or fully-developed. In order to confirm these conclusions, the distribution of IF with stability function z/L during the nighttime is given in Figure 1 as well. Under strong stable conditions (i.e. z/L » 1), turbulence is weak and IF is nearly zero. While the weak stable cases (i.e. z/L≈0.1) are accompanied by active turbulence but larger negative IF. This part is added to the revision as in: "Fig. 7 further confirms the relationship between IF and u_* or z/L, in which dots of strong turbulence (u_*) and weak stable stratification (z/L≈0.1) mainly come from the TS. Larger deviation of IF occurs accompanied by increasing turbulent strength when stability in the ABL becomes weaker. That is, intermittent turbulence (marked by large negative values of IF) leads

to strong fluxes during the TS." (page 13 lines 7-10) and Fig.7 (page 14).

(3) Your measurements are from Tianjin, which is just west of the Bohai Bay. The emission and formation of PM2.5 over the land areas are much stronger than over the sea. Therefore, I guess air from the Bohai Bay was much cleaner. During each TS the prevailing wind was either southeasterly or northeasterly, different from that during the CSs. Did the change in horizontal air flow contribute also to the decrease in the PM2.5 concentration? And how significant?

Response: We really appreciate your constructive questions. Firstly, we must apologize that there is something wrong with the wind vector in Fig. 2. We have carefully checked through the raw data and corrected the drawing program. The right wind vector at three levels (40, 120, and 200 m) for two cases is shown in the following Figure 2 and Figure 3 is the corresponding rose diagram of wind direction. From the results of Figure 2, the CS is characterized by south-easterly wind. When it comes to the TS, the wind predominately originates from west or north-west. Although the wind vector of Case-2 in Figure 2 is relatively disordered, the rose diagram in Figure 3 reveals that the most common wind direction for the CS ranges from east to south-east and the flow during the TS is mainly from the west, which is consist with the results of Case-1. Previous works have given plenty of solid evidence on the impacts of local and synoptic circulation on the accumulation and transport of air pollutants in the North China Plain. Considering that the main purpose of this work focuses on the vertical transport by turbulent mixing, we just cited some recently published paper (including Zhang et al., 2017; Miao et al., 2017; Ye et al., 2016; Zhang et al., 2012; Zheng et al., 2015a; Jiang et al., 2015) in the introduction (see page 2 line 4). Here we try to address it in detail. Tianjin is surrounded by Hebei province and also located to the north of western Shandong and northern Henan province, which are the most densely pullulated regions with the fastest growing economy in Northern China recently (Wang et al., 2014). In this case, the contribution to the PM concentration by cross-city transport from neighboring province cannot be neglected. Using modeling study, Jiang et al. (2015) revealed

that the southerly wind at lower layer contributes to transport PM from the southern neighboring cities with serious pollution. Furthermore, air masses from the south are warmer and wetter than the northern air masses, thus possessing a higher specific humidity, which facilitates the secondary formation by heterogeneous reactions (Zheng et al., 2015a). In terms of transport stage, Zheng et al., (2015a) found that weather pattern for the clean hours are normally characterized by strong high-pressure centers northwest of the polluted region in winter (i.e., the Siberian anticyclone), resulting in strong north-westerly wind. We are also thankful for your revealing comment on the sea breeze. It is reasonable to expect that the flows from the Bohai sea would be helpful to improve the air quality in Tianjin, but the work by Miao et al. (2015) provides an opposite conclusion. Their modeling results show that the southerly ambient wind brings lots of aerosols emitted from southern region to the Bohai sea and then sea breeze transports marine air together with the aerosols to the land. All of these works above present the importance of horizontal circulation in the transport of PM2.5. However, some works (Zheng et al., 2015a; Chan and Yao, 2008) mentioned that in the case of city clusters, air pollution may not be eliminated solely by advection. This is because the pollution is formed in the city cluster, there is no clean air from upwind, resulting in more persistent pollution events. There is no doubt that horizontal transport is crucial to the decrease in the PM2.5 concentration, but the mechanisms of polluted events are complicated and we try to explore the effect of intermittent turbulence from a new angle. We are so sorry for the errors in the wind vector in Fig. 2 and they have been corrected in the revision. Besides, in order to enrich this work, some previous results on the local circulation are introduced in page 7 lines 14-22: "For Case-1, wind at lower levels mainly comes from the south-east during the CS, while the dominant wind direction turns into west when it comes to the TS. Although the wind direction for Case-2 is seemingly unsteady in Fig. 2, the statistical the rose diagrams (see Figure S8) confirm a similar result, with south-easterly flows dominating the CS and westers for the TS. This wind-direction pattern is in agreement with previous works (Zhang et al., 2017; Miao et al., 2017; Zheng et al., 2015a; Jiang et al., 2015). They found that

south-easterly wind can bring the aerosols emitted by the surrounding cities to this region while the clean hours is normally characterized by strong high-pressure centers northwest of the polluted region in winter. However, in the region with densely distributed mega-cities (as in the case of Tianjin), because the upwind flows is polluted, mere advection may not be enough to disperse pollutants, thus resulting in persistent air pollution events (Zheng et al., 2015a; Chan and Yao, 2008)."

Figure 2 Wind vector at three levels. The left panel is for Case-1 and the right panel is for Case-2.

Figure 3 Comparison of rose diagram between the CS and the TS for two cases at three levels.

(4) You show in Fig. 3 vertical profiles of changes in potential temperature during the CSs. I think similar results for the TSs should be presented and discussed as well. You may show how rapidly the stable condition formed during the CSs was broken by intermittent turbulence. In addition, some discussions about the evolution of the PBL height may be also good for a more complete picture.

Response: Thank you so much for your constructive advice. The change in potential temperature during the CS and the TS is both presented in Fig.3 (a–f) in the reversion. Compared with the clear warming at high levels during CSs, the high-level cooling during TSs is significant (see Fig. 3 d–f), implying the collapse of inversion layer when it comes to the TS. Meanwhile, Fig. 3g gives the daily mean potential temperature profiles from 23 to 28 November to illustrate the evolution of inversion layer. We can see that the inversion layer gradually developed from 23 to 27 November but rapidly collapsed on 28 November. Since there is no radio soundings available near Tianjin site, we simulated the Planetary Boundary Layer Height using WRF model. The model configuration can refer to the work by Zheng et al. (2015c) which focused on a haze event in 2013 of this region. Details are given as follows: "Fig. 3 depicts the distribution of Planetary Boundary Layer Height (PBLH) and the daily mean potential temperature

profiles at 15 different heights, including change of $\theta$ over the CS (Fig. 3a–c) / TS (Fig. 3d–f) and the development during the whole polluted event (Fig. 3g). The $\Delta\theta$ at given height of CSs was calculated by subtracting the value of $\theta$ on the last day from that on the first day. And so it does for TSs. For Case-1, $\Delta\theta$ during the CS at the lowest level (5 m) is only 5.2 K. But for the top level at 250 m, $\Delta\theta$ is relatively larger with a value of 6.8 K. This result confirms that the warming of upper layers is stronger than that of lower layers, implying an increasingly stably stratified boundary layer during polluted days. Figs. 3b and 3c for Case-2 verify this conclusion as well. On the contrary, $\Delta\theta$ during TSs (Fig. 3d–f) presents a significant cooling at higher levels, denoting the collapse of inversion layer at the end of the polluted event. Taking Case-1 as an example, Fig. 3g depicts the evolution of inversion layer. It can be seen that the inversion layer was gradually enhanced from 23 to 27 November but quickly depressed on 28 November, which verifies the results of Fig. 3a–f. Fig. 3h illustrates the distribution of PBLH, which is simulated with the Weather Research & Forecasting (WRF) Model (Zheng et al., 2015c). In Fig. 3h, the PBLH for Case-1 gradually decreased and reached its minimum on the night of 26–27 November. Then the PBLH redeveloped to higher than 1,300 m during the daytime of 28 November." (page 7 lines 29-33 and page 8 lines 1-9)

Figure 4 Vertical distribution of daily mean potential temperature. The change of daily mean potential temperature of CSs is showed in (a)–(c) and (d)–(f) are for TSs. (g) illustrates the evolution of inversion layer of Case-1. (h) is the PBLH simulated with WRF Model.

(5) Your results and conclusions are based on cases study. I think it is better to add "cases from Tianjin" or similar subtitle. And "vertical diffusion" in the title can be questionable if you cannot prove that the decrease in PM2.5 was solely due to the vertical diffusion.

Response: Thank you for your suggestion. We specified "cases from Tianjin" in the revision. But considering that there have been a lot of work aiming on the horizontal transport of PM2.5, we mainly focus on the effect of vertical mixing of intermittent

turbulence. Indeed, the reasons for the transport of particles are complicated, including climate change, synoptic circulation, and boundary layer structures and we cannot address them all. So far, there is limited works on the intermittent turbulence under strongly stable conditions. Therefore, we keep "vertical" in the new title to emphasize the effects of vertical turbulent mixing and we hope you will approve of our modification. The new title is "Intermittent turbulence contributes to vertical dispersion of PM2.5 in the North China Plain: cases from Tianjin".

Minor points: Page 1 line 21: What do you mean by "wind filed"? Wind profile?

Response: Yes, it should be "wind profile" and has been rewritten.

Page 2 line 21: Define "FI".

Response: "FI" is defined as flux intermittency by Mahrt (1998). FI = $\sigma$F/abs[F], in which $\sigma$F is the standard deviation of the 5-minute averaged flux and abs[F] is the absolute value of the one-hour average of the flux. The manuscript has been corrected as well. "FI index (Flux Intermittency, Eq. (9) in Mahrt, 1998)" (page 2, lines 21-22)

Page 2 line 32: Delete "respectively".

Response: Yes.

Page 3 line 4: Change "east" to "southeast".

Response: Yes.

Page 3 line 11: I think "HMP45C" is the type name of the probe and should be put in the brackets.

Response: This has been corrected.

Page 3 line 16: Was the TEOM system installed near the tower or the WPR? Please make it clear.

Response: Thank you for your question. The TEOM system used in this study is

mounted near the 255-m tower. The distance between the 255-m tower and the TEOM system is around 2.3 km. The location of the TEOM system is specified as follows: "The 1405-DF TEOM system is located nearly 2.3 km away from the 255-m tower to the east and installed at a height of 3 m to monitor the surface PM2.5." (page 3 lines 15-17)

Table 1: "c: 300-366 m s-1". Is this the range of wind speed that the sonic anemometer can measure? 300 is a very strange number here.

Response: Here c represents the speed of sound which is used to calculate the sonic virtual temperature. According to the instruction manual of CSAT3 Three Dimensional Sonic Anemometer, the range of speed of sound is from 300 to 366 m s–1 (–50 to +60 °C). The definition of c is added to Table 1.

Page 4 line 16: Change "poor data" to "poor quality of data".

Response: Yes, thank you.

Page 4 lines 19-21: What are the criteria for data that are suitable for this study?

Response: All of the data used in this study were checked strictly. The quality control for turbulence observations includes error flag, spike detection, cross wind correction, spectral loss correction, sonic virtual temperature correction, density fluctuation correction, and coordinate rotation. "If more than 20% points within a given 30-min time series were detected as outliers, then this 30-min observation was discarded." (page 4 lines 10-11) The wind profiles were checked time by time. "First, data below 200 m were removed due to the interference of surrounding environment, including trees and buildings. Then each vertical profile was checked through and points with larger than 2.5 standard deviations were regarded as outliers and discarded. (page 4 lines 17-19)" And according to previous study in this region (Wei et al., 2014), a profile was discarded if more than 40% of the data points were outliers or missing.

Page 5 line 2: "local standard time" or "Beijing Time"?

Response: Yes, it should be "Beijing Time".

Page 5 line 9: "On this basis"? It is not clear what is denoted.

Response: We mean that based on the arbitrary-order HSA, we developed IF. This has been corrected.

Page 5 lines 11-12: Delete "(CSAT3, CAMPBELL Inc., USA)" because the same information is given on page 3.

Response: It has been deleted.

Page 5 line 19: Do you mean the local maxima that are found within every 30-min periods?

Response: Yes, here the local maxima are from the 30-min time series X(t). This has been corrected as: "The first step is to form the upper envelope e_max (t) based on the local maxima of 30-min X(t)" (page 5 lines 21).

Page 5 line 9, page 6 lines 12-13, and page 11 lines 18-19: You are proposing or defining IF at these three places. This is redundant. I think you should define the IF index at a suitable place and use it elsewhere.

Response: Thank you for pointing out that. The IF index is defined when it is first mentioned (page 5 line 11) and other definitions have been deleted.

Page 6 line 31 and page 7 line 1: ". . .increased to a maximum of 412 ug m-3 for PM2.5 and then dropped to a low level within a few hours no matter for Case-1 and Case-2". Please check you expression. I do believe the maximum values in Fig. 2a and Fig. 2b are all 412.

Response: Thank you for your question. The maximum for Case-1 is 263 $\mu$g m-3 and 412 $\mu$g m-3 for Case-2. This part has been rewritten in the revision. "it can be seen that the concentration of PM2.5 gradually increased to maxima (263$\mu$g m-3 for Case-1 and 412 $\mu$g m-3 for Case-2) and then dropped to a low level within a few hours." (page

7 lines 5-6)

Page 9 Fig. 2: Please add some ticks on the Y-axes of Figs. 2a and 2b.

Response: The Fig. 2a and 2b have been replotted with ticks.

Page 11 line 2: CSs or TSs?

Response: We are sorry for the slip of the pen. It should be "TSs".

Page 11 Fig. 5: Does each concave curve represent a 30-min result? Please make it clear.

Response: Yes, each curve in Fig. 5 is from a 30-min vertical wind speed signal, which has been clarified in the caption of Fig. 5. "Hilbert-based scaling exponent function at 40 m during different stages for (a) – (b) Case-1 and (c) – (d) Case-2, where each dashed curve represents the result of 30-min vertical wind speed signal and the black solid line denotes the K41 result q/3." (page 12 and lines 15-16)

Page 11 line 24: Delete "site".

Response: Yes.

Page 13 line 2: "under stable conditions"? Are you not talking about the TS?

Response: Thank you for your question. We have replaced "under stable conditions" with "in the ABL".

References

Huang, Y. X., Schmitt, F. G., Lu, Z. M., & Liu, Y. L. (2008). An amplitude-frequency study of turbulent scaling intermittency using empirical mode decomposition and Hilbert spectral analysis. EPL (Europhysics Letters), 84(4), 40010.

Wei, W., Zhang, H. S., Schmitt, F. G., Huang, Y. X., Cai, X. H., Song, Y., ... & Zhang, H. (2017). Investigation of Turbulence behaviour in the stable boundary layer using arbitrary-order Hilbert spectra. Boundary-Layer Meteorology, 163(2), 311-326.

Chan, C. K., & Yao, X. (2008). Air pollution in mega cities in China. Atmospheric environment, 42(1), 1-42.

Jiang, C., Wang, H., Zhao, T., Li, T., & Che, H. (2015). Modeling study of PM 2.5 pollutant transport across cities in China's Jing–Jin–Ji region during a severe haze episode in December 2013. Atmospheric Chemistry and Physics, 15(10), 5803-5814.

Miao, Y., Guo, J., Liu, S., Liu, H., Zhang, G., Yan, Y., & He, J. (2017). Relay transport of aerosols to Beijing-Tianjin-Hebei region by multi-scale atmospheric circulations. Atmospheric Environment, 165, 35-45.

Wang, H., Tan, S. C., Wang, Y., Jiang, C., Shi, G. Y., Zhang, M. X., & Che, H. Z. (2014). A multisource observation study of the severe prolonged regional haze episode over eastern China in January 2013. Atmospheric Environment, 89, 807-815.

Wei, W., Zhang, H. S., & Ye, X. X. (2014). Comparison of low‐level jets along the north coast of China in summer. Journal of Geophysical Research: Atmospheres, 119(16), 9692-9706.

Zheng, G. J., Duan, F. K., Su, H., Ma, Y. L., Cheng, Y., Zheng, B., ... & Pöschl, U. (2015a). Exploring the severe winter haze in Beijing: the impact of synoptic weather, regional transport and heterogeneous reactions. Atmospheric Chemistry and Physics, 15(6), 2969-2983.

Zheng, B., Zhang, Q., Zhang, Y., He, K. B., Wang, K., Zheng, G. J., ... & Kimoto, T. (2015c). Heterogeneous chemistry: a mechanism missing in current models to explain secondary inorganic aerosol formation during the January 2013 haze episode in North China. Atmospheric Chemistry and Physics, 15(4), 2031.

Please also note the supplement to this comment:
https://www.atmos-chem-phys-discuss.net/acp-2018-121/acp-2018-121-AC1-supplement.pdf

[Figure]

[Figure]

[Figure]

**Fig. 1.** Scatter plot comparing IF and other variables (u_* and nighttime z/L) for two cases at 40 m. The dashed lines are the fittings from least-squares regression and the triangle marks the cross point.

[Figure]

**Fig. 2.** Wind vector at three levels. The left panel is for Case-1 and the right panel is for Case-2.

[Figure]

**Fig. 3.** Comparison of rose diagram between the CS and the TS for two cases at three levels.

**Fig. 4.** Vertical distribution of daily mean potential temperature. The change of daily mean potential temperature of CSs is showed in (a)–(c) and (d)–(f) are for TSs. (g) illustrates the evolution of inversio

[Figure]

**Fig. 5.** Figure 2 in the manuscript. Time series of surface PM2.5, wind vector, temperature (T), relative humidity (RH), horizontal wind speed (U), vertical wind speed (W), friction velocity (u_*), turbulent

---

## Author Comment (AC2) · 4 Jul 2018

Dear reviewer, We really appreciate your revealing questions and comments. Some key points about the intermittent turbulence in the SBL that you have brought up with are helpful to improve this work. We also thankful for the useful references you suggested. We did our best to respond to these comments one by one. We hope the reviewer would approve of our following response.

Normally, intermittent turbulence is associated with the stable boundary layer (SBL). That is, turbulence strength varies when the background stratification is generally stable. When the stratification is totally wiped out by strong turbulent mixing for a relatively long period, the relatively strong turbulent mixing is not considered as part of a time se-

ries of intermittent turbulent mixing anymore. In this study, a period of strong turbulent mixing occurred at the end of all the three cases and each of them lasted for nearly a day. If the authors think the strong turbulent mixing period in each case is part of the time series of intermittent turbulence, this is definitely not what intermittent turbulence in the traditional definition.

Response: We really appreciate your questions. As your comments note, intermittent turbulence in the SBL is manifested by sporadic bursts lasting from tens of seconds to several minutes. In this study, turbulence during TSs is much stronger than that of CSs and after checking through all of the raw time series, we find that vertical wind speed during TSs is characterized by episodic and intermittent events. The primary reason of the seemingly continuous turbulence during TSs in Fig.2 is that the 10-Hz observed turbulence is confined to a narrow plot and the weak events are covered by the relatively stronger fluctuations, which makes it look like continuous. For space reasons, here we take three examples to illustrate the details of intermittent turbulence during the TSs, one for Case-1 and the other two for Case-2. Meanwhile, the intrinsic mode functions (IMFs) from the empirical mode decomposition are given. As shown in the following Figure 1, the raw time series of vertical wind speed are not fully-developed and continuous. On the contrary, the strength of turbulence is variable. The stronger turbulence can last several minutes and frequently happens within each time series (marked by shaded areas). And this characteristic is more obvious in the high frequent IMFs (i.e. from IMF1 to IMF8). These results show that the relatively stronger turbulence during TSs happens intermittently but not continuously. Some previous observations also confirmed that intermittent bursts of turbulence and mixing can also occur multiple times (Poulos, et al., 2002). We suppose that the intermittent turbulence during TSs in Fig. 2 is just covered by the frequently happened bursts.

Figure 1 Three examples of IMFs and 30-min vertical wind speed from TSs. The shaded areas mark the relatively strong intermittent "burst".

Meanwhile, we use some other indexes, including kurtosis and FI (Flux Intermittency
defined by Mahrt, 1998), to verify the behavior of intermittency. The normalized probability density function of fully-developed turbulence should be Gaussian but intermittency would modify its shape. Therefore, kurtosis (K=(ãĂŰu'ãĂŮ^4 ) ÌĚ/$\sigma$^4) that characterizes the variation of probability distribution could be introduced as an intermittency index (Vindel et al., 2008). Another useful index for intermittency is FI, which is shorthand for flux intermittency proposed by Mahrt (1998) and has been applied in many works (e.g. Ha et al., 2007). FI is defined as $\sigma$_F/|F|, where $\sigma$_F is the standard deviation of the averaged friction velocity and |F| is the absolute value of friction velocity. Considering the limited scale in the SBL, $\sigma$_F is based on 1-min values of friction velocity and |F| is 30-min averaged. Figure 2 presents Kurtosis and FI, compared with simultaneous PM2.5 concentration and IF. It can be seen that the values of Kurtosis and FI are much larger during TSs, which is consistent with the development of PM2.5 concentration and IF values. All of the parameters confirm that, the relatively stronger turbulence during TSs is intermittent but not fully-developed. Besides, it should be noticed that the small values of turbulence during CSs are mainly due to the extremely weak turbulent fluctuation. Considering that the objective of this work is not the comparison between different methods, we just applied the arbitrary-order HSA technique into this study. According to Huang et al. (1998), this method is intuitive, direct, and adaptive, with a posteriori-defined basis, from the decomposition method, based on and derived from the data, which makes it suitable for the analysis of nonlinear and non-stationary turbulence signals in the ABL. Detailed discussion on the methodology is given in the supplement. The results of Kurtosis and FI have been added to verify the conclusion as in: "In order to validate the results of IF, another two parameters to indicate the intermittency of turbulence were developed using the same data: one is kurtosis (Vindel et al., 2008) and the other is FI (Mahrt, 1998; Ha et al., 2007). The results of both kurtosis and FI are consistent with those of IF (see Figure S7)." (page 13, line25, lines 3-6)

Figure 2 Comparison of (a) – (b) PM2.5 concentration, (c) – (d) IF, (e) – (f) Kurtosis, and (g) – (h) FI by Mahrt (1998). Left panel is for Case-1 and right panel is for Case-2.

Physically, intermittent turbulent mixing in the SBL can be generated by intermittent strengthening wind shear, which can be associated with low level jets (LLJs) as pointed by the authors in the paper. However, this mechanism is not new; Banta et al. (2003, JAS; 2006, QJ; 2007 JAS) have investigated relationships between LLJs and turbulent mixing extensively.

Response: Yes, a series of works have focused on the relationship between the LLJs and intermittent turbulence, which is a widely-accepted mechanism. Based on this mechanism, we further attempted to reveal the effect of intermittent turbulent mixing on the dispersion of PM2.5 from a viewpoint of small-scale turbulent structure. As far as we know, there is few works aiming on this topic and we hope this study could provide a different new angle to think about the possible reasons for the dispersion of near surface PM2.5. We are also thankful for your useful references. These works enhanced our understanding on the LLJs and intermittent turbulence and we cited these works in the revision. "The reasons for intermittent turbulence in the ABL have not yet been well understood. Some potential causes include gravity waves (Sorbjan and Czerwinska, 2013; Strang and Fernado, 2001), solitary waves (Terradellas et al., 2005), horizontal meandering of the mean wind field (Anfossi et al., 2005), and low-level jets (LLJs, Marht, 2014; Banta et al., 2007; 2006; 2003)." (page 15, lines 2-5)

In addition, from the title of the paper, it seems that the authors would address the role of intermittent turbulent mixing to the vertical dispersion (not diffusion, diffusion is for molecular movements) of PM2.5. However, the observation indicates that the intermittent turbulent mixing during the high PM2.5 period is not strong enough to disperse PM2.5 and the significant reduction of PM2.5 is observed at the end of each event when strong mixing arrives. Then what is the significance of intermittent turbulence during the stable period? What is the significance of the new intermittent turbulence index introduced here in comparison with simple parameters such as wind speed if wind shear is the key physical process for dispersing PM2.5?

Response: Thank you so much for your comments. The title is changed into "Intermittent turbulence contributes to vertical dispersion of PM2.5 in the North China Plain: cases from Tianjin" in the revision. And we have checked through the paper to rewrite "diffusion" with "dispersion". We are sorry for the ambiguity. Actually, the turbulence during TSs belongs to intermittent regime while the CS can be considered as a more stable regime and the turbulence is largely suppressed. The turbulence during CSs is too weak with mean u* and TKE less than 0.3 m/s and 0.5 m2/s2 respectively, and z/L during the nighttime is much larger than 1, which could be sorted as the extremely stable regime or radiation regime (as in Mahrt, 2014). On the other hand, the turbulence of the TS is relatively stronger but not strong enough. These are two totally different stratification conditions. In order to reveal the distinction of different stages and the corresponding turbulence, an index (IF) was proposed. According the theory of arbitrary-order HSA, the larger deviation of scaling exponent $\xi(q)$ represents stronger intermittency of turbulence. At this point, the turbulence during TSs is intermittent, according to the distribution of IF (Fig.6). On the contrary, the values of IF during CSs are near zero, which is mainly attributed to extremely weak fluctuation (u_* less than 0.3 m/s). To avoid ambiguity, the time with u_*<0.3 m/s is colored as grey in Fig.6. As mentioned in the first response above, the results of other indexes (i.e. kurtosis and FI) also confirm the conclusion in this study, that is, the turbulence during TSs is intermittent and the associated intermittent turbulent mixing facilitates the dispersion of PM2.5 near the surface. We emphasized this in the revision: "The results show that the turbulence is very weak during the cumulative stage due to the suppression by strongly stratified layers; while for the stage of dispersion, the turbulence is highly intermittent and not locally generated." (page 1 lines 20-22) "Any of these mechanisms would destroy the statistical symmetries stored in the fully developed turbulence, resulting in deviations from K41's q/3 and a set of concave curves in which the degree of the discrepancy of concave curves manifests the strength of turbulent intermittency." (page 11 lines 21-22 and page 12 lines1-2) "The Hilbert-based exponent scaling function $\xi(q)$ shows great deviations from K41's theoretical result of q/3 by a set of concave curves, indicating that the enhanced turbulence in the ABL when entering the TS is intermittent rather

than continuous or fully developed." (page 17 lines 18-20)

Furthermore, because the stable boundary layer is known to be associated with weak winds when wind direction variations can be significant, how does wind advection contribute to the temporal variation of PM2.5 besides vertical dispersion of PM2.5 by turbulent mixing? Where is the high PM2.5 source?

Response: Thank you for your questions. PM2.5 pollution in Tianjin and its neighboring cities (i.e., Beijing) has received great attention and a series of works have studied the impacts of synoptic and local circulation (Zhang et al., 2017; Miao et al., 2017; Ye et al., 2016; Zhang et al., 2012; Zheng et al., 2015a; Jiang et al., 2015). Here we summary their main conclusions. Tianjin is located in one of the most polluted city clusters (the so-called Beijing-Tianjin-Hebei region, BTH region) and surrounded by Hebei, western Shandong and northern Henan, several most severely polluted provinces in Northern China (see map in Figure 3). Therefore, southerly flows from the polluted areas would deteriorate the air quality in Tianjin (Zhang et al., 2017; Miao et al., 2017; Zheng et al., 2015a; Jiang et al., 2015). Tianjin features a four-season climate and is under the influence of the Siberian anticyclone in winter. The strong north-westerly wind from the Siberian anticyclone in winter is helpful to the advection of air pollutants. Therefore, the clean days are associated with the high-pressure centers northwest of the polluted region. In order to summarize the circulation mechanism, Fig.2 also illustrates the wind vector during these two cases. Please see page 7 line 14-20: "For Case-1, wind at lower levels mainly comes from the south-east during the CS, while the dominant wind direction turns into west when it comes to the TS. Although the wind direction for Case-2 is seemingly unsteady in Fig. 2, the statistical the rose diagrams (see Figure S8) confirm a similar result, with south-easterly flows dominating the CS and westers for the TS. This wind-direction pattern is in agreement with previous works (Zhang et al., 2017; Miao et al., 2017; Zheng et al., 2015a; Jiang et al., 2015). They found that south-easterly wind can bring the aerosols emitted by the surrounding cities to this region while the clean hours are normally characterized by strong high-pressure centers

[Figure]

northwest of the polluted region in winter." Some works (Zhang et al., 2017; Wang et al., 2014) have revealed that reginal transport and local emission both play an important role in air quality in the BTH region. One work by Zhang et al., (2017) focusing on the primary PM2.5 found that in Tianjin, primary PM2.5 mainly originated from local emission before heavy pollution events; when it comes to polluted periods, the contribution from non-local region increased and amount of pollutants were transported from Shandong, Henan, even Jiangsu and Anhui via the low-level southerly flows. In addition, some works (Zheng et al., 2015a; Chan and Yao, 2008) have pointed out that if a large-area pollution event occurred in densely distributed mega-cities (as in BTH region), air pollution might not be eliminated solely by advection considering that the up-wind flows are polluted as well. Based on these solid results by previous works, this work tries to reveal the effects of the vertical transport of intermittent turbulence which is a relatively new and different view so far. "However, in the region with densely distributed mega-cities (as in the case of Tianjin), because the upwind flows is polluted, mere advection may not be enough to disperse pollutants, thus resulting in persistent air pollution events (Zheng et al., 2015a; Chan and Yao, 2008)." (page 7 lines 20-22)

Figure 3 Map of Tianjin and its surrounding cities.

References

Chan, C. K., & Yao, X. (2008). Air pollution in mega cities in China. Atmospheric environment, 42(1), 1-42.

Ha, K. J., Hyun, Y. K., Oh, H. M., Kim, K. E., & Mahrt, L. (2007). Evaluation of boundary layer similarity theory for stable conditions in CASES-99. Monthly Weather Review, 135(10), 3474-3483.

Huang, N. E., Shen, Z., Long, S. R., Wu, M. C., Shih, H. H., Zheng, Q., ... & Liu, H. H. (1998). The empirical mode decomposition and the Hilbert spectrum for nonlinear and non-stationary time series analysis. In Proceedings of the Royal Society of London A: mathematical, physical and engineering sciences, 454(1971), 903-995.

Mahrt, L. (1998). Nocturnal boundary-layer regimes. Boundary-layer meteorology, 88(2), 255-278.

Mahrt, L. (2014). Stably stratified atmospheric boundary layers. Annual Review of Fluid Mechanics, 46, 23-45.

Poulos, G. S., Blumen, W., Fritts, D. C., Lundquist, J. K., Sun, J., Burns, S. P., ... & Terradellas, E. (2002). CASES-99: A comprehensive investigation of the stable nocturnal boundary layer. Bulletin of the American Meteorological Society, 83(4), 555-581.

Vindel, J. M., & Yagüe, C. (2011). Intermittency of turbulence in the atmospheric boundary layer: Scaling exponents and stratification influence. Boundary-layer meteorology, 140(1), 73-85.

Wang, Z., Li, J., Wang, Z., Yang, W., Tang, X., Ge, B., ... & Wand, W. (2014). Modeling study of regional severe hazes over mid-eastern China in January 2013 and its implications on pollution prevention and control. Science China Earth Sciences, 57(1), 3-13.

Zhang, Y., Zhu, B., Gao, J., Kang, H., Yang, P., Wang, L., & Zhang, J. (2017). The source apportionment of primary PM2. 5 in an aerosol pollution event over Beijing-Tianjin-Hebei region using WRF-Chem, China. Aerosol and Air Quality Research, 17, 2966-2980.

Zheng, G. J., Duan, F. K., Su, H., Ma, Y. L., Cheng, Y., Zheng, B., ... & Pöschl, U. (2015). Exploring the severe winter haze in Beijing: the impact of synoptic weather, regional transport and heterogeneous reactions. Atmospheric Chemistry and Physics, 15(6), 2969-2983.

Zheng, G. J., Duan, F. K., Su, H., Ma, Y. L., Cheng, Y., Zheng, B., Zhang, Q., Huang, T., Kimoto, T., Chang, D., Pöschl, U., Cheng, Y. F. and He, K. B.: Exploring the severe winter haze in Beijing: The impact of synoptic weather, regional transport and

heterogeneous reactions, Atmos. Chem. Phys., 15(6), 2969–2983, doi:10.5194/acp-15-2969-2015, 2015a.

Please also note the supplement to this comment:
https://www.atmos-chem-phys-discuss.net/acp-2018-121/acp-2018-121-AC2-supplement.pdf

———————————————————

[Figure]

[Figure]

**Fig. 1.** Three examples of IMFs and 30-min vertical wind speed from TSs. The shaded areas mark the relatively strong intermittent "burst".

[Figure]

**Fig. 2.** Comparison of (a) – (b) PM2.5 concentration, (c) – (d) IF, (e) – (f) Kurtosis, and (g) – (h) FI by Mahrt (1998). Left panel is for Case-1 and right panel is for Case-2.

[Figure]

**Fig. 3.** Map of Tianjin and its surrounding cities.

---

## Author Comment (AC3) · 4 Jul 2018

Dear reviewer, Thank you so much for all these recommendations. The manuscript has been refined following your advice. In order to enrich this work, some more detailed information on the methodology is given in a supplement due to space limit. The detailed responses are as follows:

This manuscript investigates the role of intermittent turbulence in alleviating heavy pollution episodes that frequently occur in China. The papers includes a theoretical background and analysis of measurement data related to 2 pollution episodes. While the vast majority of the prior research on air pollution episodes in China has concentrated on the factors favoring the accumulation of pollutants, this paper investigates a phenomenon that helps to get rid of high pollution levels. As such, I think that this paper is original enough to warrant publication in ACP. I have a few issues that should, however, be addressed before the publication.

The authors introduce an Intermittent Factor (IF) which they use for explaining the effects of intermittent turbulence on the observations. I have a few comments on this. First, it seems that q is the key variable when determing IF. Therefore, the authors should explain more explicitly what is the exact meaning of q, not just to mention that it is the power exponent of the instantaneous amplitude of something. Second, IF is defined such that it is zero for fully developed turbulence and negative if not. However, the exact value of IF does not tell anything for the reader. Would it be possible to provide some idea how to interprete the value of IF. How small (or large in absolute sence) should IF be for the intermittent turbulence to be important etc?

Response: We all appreciate your constructive comments. A sketch of the arbitrary-order HSA has been given in the manuscript, including the derivative process, equations and a brief comparison with existing methods, (i.e., Fourier analysis and wavelet transform). Due to lack of space, we cannot address the method and the relative parameters fully in the manuscript, so we write a supplement (Figure S1-S6 and the relative illustration) to describe the process of the arbitrary-order HSA. As you pointed out, q is an important parameter in the derivation of scaling exponent function $\xi(q)$ for several reasons. Firstly, q is the moment of the arbitrary-order Hilbert spectrum $L\_q\ (\omega)=\int p(\omega,A)A\char`^q\ dA$. If q is taken as 2, the second-order Hilbert spectrum $L\_2\ (\omega)=\int p(\omega,A)A\char`^2\ dA$ can be an analogical representation of classic Fourier energy spectrum, given that the square of amplitude is equivalent to energy density. Secondly, in the identification of the range of scale invariance, the third-order Hilbert spectrum $L\_3\ (\omega)=\int p(\omega,A)A\char`^3\ dA$ is taken as the reference. Kolmogorov's initial proposal leads to $\zeta(q)=q/3$ (Kolmogorov, 1941), while the scaling exponent function $\zeta(q)$ of intermittent turbulence is nonlinear and concave. Only $\zeta(3)$ has no intermittency correction, that is, $\zeta(3)=1$. Finally, the highest moment of $\zeta(q)$ considers both computing efficiency

and accuracy in the measurements of high-order moments. The higher the order is, the longer the length of sample needs, while the arbitrary-order HSA process of long data will take a lot of time. Therefore, the maximal moment is taken as $q\_max=4$. The details on the method have been elaborated in the supplement. In terms of the values of IF, the magnitude of IF changes from different observation sites based on our past experience. And from Fig. 6 in the manuscript, it can be seen that the absolute values of IF increase with higher levels, which implies that there is no universal criteria for the identification of intermittency using IF. However, we can extract a reference value for these two cases in Tianjin in this study. The following figure (also Fig.7 in the revision) illustrates the relationship between IF and $u\_*$ and nocturnal $z/L$. The absolute IF increases with stronger turbulent strength when stability in the ABL becomes weaker. Hence, a point of intersection can be identified using the fittings from least-squares regression and represent a critical IF value at given level (here 40 m) beyond which the intermittency of turbulence in the ABL is significant. Based on the regression analyses, the cross points correspond to IF values of -0.53 (Case-1) and -0.50 (Case-2), respectively. Hence, in the present study, we adopt -0.5 as a threshold for the strong intermittency in the ABL. Some relative discussion has added to the revision. "Besides, the points of intersection from the least-squares regression in Fig. 7 could denote the threshold beyond which the intermittency of turbulence arises under the mutual influence of dynamic and thermodynamic. The values of IF are -0.52 and -0.50 for Case-1 and Case-2, respectively. Hence, a cut-off value of IF (-0.50) can be identified to manifest the significant intermittency of turbulence. But it should be kept in mind that the absolute values of IF change from different heights and sites and this cut-off value of IF can only be used as a reference in the present study." (page 13 lines 12-16) "For 40 m, a cut-off value of IF (-0.50) indicates the initiation of strong turbulent intermittency in the ABL, while this is not a universal value and the threshold varies with different cases." (page 18 lines 4-6)

Figure 7 in the manuscript. Scatter plot of IF vs. $u\_*$ and $z/L$ (night time) for (a) Case-1 and (b) Case-2 at 40 m. The dashed lines are the fittings from least-squares regression

and the triangle marks the cross point.

The discussion of Figure 5 in the beginning of page 11 is a bit confusing. The authors state that the difference for CSs is much more obvious. I do not understand this statement. By looking at the figure, the differ curves for TS show more spread than the curves for CS. So what are the authors referring to when discussion about differences? Also, figures 5a-d have the straight line for fully developed turbulence (faint solid black lines). This line should show up more clearly in the figure and it should be said that it is a solid line.

Response: Thank you so much for your detailed comments. Yes, the beginning of page 11 should be "TSs". We apologize for this slip of the pen and have corrected it in the revision. Please see page 10 lines 3-4: "However, the difference for TSs is much more obvious (in Figs. 5b and 5d), indicating stronger intermittency in the turbulence." Also, bolder lines for q/3 are used in Fig.5 and the caption has been written as "the black solid line".

Figure 5 in the manuscript. Hilbert-based scaling exponent function at 40 m during different stages for (a) – (b) Case-1 and (c) – (d) Case-2, where each dashed curve represents the result of 30-min vertical wind speed signal and the black solid line denotes the K41 result q/3. (e) compares vertical wind fluctuation at 40 m between the CS (before 00:00 on 26 January 2017) and TS (after 00:00 on 26 January 2017). The latter shows apparent 'bursts' marked by the rectangular frame.

Please check out that all the used mathematical symbols are explained in the text.

Response: Thank you. We have checked through the text and defined all of the mathematical symbols.

A few grammatical issues: Page 1, line 25 should read "particulate matter" Page 14, line 7: . . .we summarize. . .

Response: We are sorry for these mistakes and have been corrected them.

Please also note the supplement to this comment:
https://www.atmos-chem-phys-discuss.net/acp-2018-121/acp-2018-121-AC3-
supplement.pdf
* * *
[Figure]

**Fig. 1.** Figure 7 in the manuscript. Scatter plot of IF vs. u_* and z/L (night time) for (a) Case-1 and (b) Case-2 at 40 m. The dashed lines are the fittings from least-squares regression and the triangle mark

[Figure]

**Fig. 2.** Figure 5 in the manuscript. Hilbert-based scaling exponent function at 40 m during different stages for (a) – (b) Case-1 and (c) – (d) Case-2, where each dashed curve represents the result of 30-min v

---

## Author Response (AR2)

**Response to Referee #3**

Dear reviewer,

   We all appreciate the useful and revealing questions you have come up with. In this response, we try our best
to answer these questions one by one. Firstly, following some previous works, we use several different methods to
category the regime of turbulence during the main PM dissipation periods, including the heat flux (i.e. Van de Wiel et
al., 2003 and many other works), the stability z/L by Mahrt et al. (1998), the threshold of minimum wind speed (Sun et
al., 2012; Van de Wiel et al., 2012), and the statistical index (Vindel et al., 2008), revealing a very stable boundary
layer and intermittent turbulence on the nights of PM dissipation. Next, we investigate the vertical transport of
turbulence, finding that the strong turbulence is generated and transported from the layers above the 255-m tower,
which verifies the role that the LLJ plays in the generation of turbulence and further validates the nature of
intermittency of turbulence in the ABL. Finally, following your comments, we have removed the part about the
radiative effects by pollutants which is irrelevant to the main topic of this work and cannot be fully proved using the
available data. The detailed explanations are as follows:

The authors responded my comments carefully. In the authors' response, they further demonstrated non-stationarity of
turbulence from their turbulent period (TS). Now I am totally confused.

The authors repeated the conventional understanding that intermittent turbulence is associated with stable conditions (P.
6, L18), but their intermittent turbulence index IF suggests large intermittent turbulence is during the weak stable
period (TS) where turbulence is strong. That is, their definition of turbulence intermittency is opposite from those in
the literature even though they claimed their intermittency index is consistent with others. It seems to me that what the
authors captured is the magnitude of w fluctuations, which should be large when turbulence is strong, but that is not
what turbulence intermittency supposes to mean. Turbulence intermittency means the magnitude of turbulent fluxes
varies significantly not individual wind components. Wind components can fluctuate significantly but turbulent fluxes
do not so that turbulent intermittency is zero. Basically what the authors show is turbulent mixing contributes to
vertical dispersion of PM2.5, strong turbulence should be associated with large u*, and weak PM2.5. This is what we
understand already. No wonder the authors claimed intermittent turbulence contributes to vertical dispersion of PM2.5.
The authors' concept of turbulent intermittency is completely wrong.

**Response:** In this part, the referee pointed out two questions. One is whether the dissipation periods with large abstract
   values of IF can be categorized as the weakly stable condition or the very stable condition. The other is the
   variable used (i.e. vertical wind speed w) to evaluate the intermittency. These two questions are related with each
   other, so we will explain them alternatively. Also, to fully address these questions, we will focus on the periods

during which the pollutant concentration changed significantly. According to the distribution of PM$_{2.5}$ concentration in Fig. 2 in the manuscript, the decrease of pollutant concentration mainly happened during the late night to the early morning. Hence, we pay our attention to the behavior of turbulence during these periods, that is, 00:00–00:06 (LS, local time) on 27, Dec, 2016 and on 26 and 29, Jan, 2017. In the following response, all of the samples are taken from 00:00–00:06 LS unless stated otherwise.

Considering the major point of this work, we need to first specify the definition of intermittency in the ABL. According the Glossary of Meteorology (Glickman 2000), the intermittency is define as "the property of turbulence within one air mass that occurs at some times and some places and does not occur at intervening times or places". In the real works, turbulence intermittency is commonly expressed as brief episodes of turbulence with intervening periods of relatively weak or unmeasurably small fluctuations (Mahrt, 1989, 1999; Van de Wiel et al., 2002; Dadic et al., 2013) or as "bursts" vividly (Mahrt, 1999; Coulter and Doran 2002; Ohya et al., 2008). In all, intermittency is an intrinsic feature of turbulence in strongly stratified flow (Mahrt, 2006) and can be manifested by a variety of variables. As the referee suggested in the previous comments, plenty of studies sought to characterize intermittency in terms of intermittent heat flux (Howell and Sun, 1999; Coulter et al., 2002; Van de Wiel et al., 2003; Doran, 2004; Steeneveld et al., 2006; Drüe et al., 2007). Meanwhile, a number of other studies tended to define turbulent intermittency by means of wind components and even temperature. For example, Sun et al. (2002) used fluctuation of vertical wind speed w and temperature T to show intermittent turbulence periods (in their Fig. 2 and 3). In the wind tunnel experiment, Ohya et al., (2008) produced the intermittent turbulence in the SBL and showed that both the horizontal/vertical wind speed and temperature captured significant turbulent intermittency. Based on the temperature observation, Lundquist (2003) verified that Hilbert-Huang transform technique is useful for exploring intermittent events in the midlatitude atmospheric boundary layer. Some other works using wind component or temperature include Muschinski et al. (2004), Salmond and Mckendry (2005) and Vindel et al. (2008), to name just a few. Sun et al. (2012) categorized different turbulence regimes with the help of turbulence strength ($\sqrt{TKE}$) and the standard deviation of horizontal/vertical wind component ($\sigma_{V/w}$) and revealed that under the influence of non-turbulence motions, the characteristics of intermittency were manifested by different variables, including horizontal/vertical wind speed, temperature and wind direction (see their Fig. 12 and 14). In the works of Reina et al. (2004) and Mahrt et al. (2013), the variance and standard deviation of vertical wind component were applied to define the onset of intermittent turbulence events. According the works summarized above, we believe that it is justified to investigate the behavior of turbulent intermittency by means of vertical wind component w in the present study.

In literature, there are mainly two kinds of methods to classify the SBL. One way uses the so-called internal system parameters (i.e. z/L, z/Λ and so on) to define the different SBL regimes. Using the stability function z/L,

Mahrt et al. (1998) suggested three regimes: a) the weakly stable regime, b) the intermediate regime with z/L > $O(0.1)$, c) the very stable regime where z/L > $O(1)$. Although the specific threshold value of z/L depends on the level of the observations, z/L under weakly stable condition should be way less than 1, that is $0 < z/L \ll 1$, to keep the continuous turbulence near the surface. From the distribution of z/L in Fig. 2 in the manuscript, it can be seen that the values of z/L during the night of 26–27 Dec, 25–26 and 28–29 Jan are much larger than $O(1)$, implying very stable case during these nights. Another thing that should be clarified here is that the small mean and standard deviation of z/L during the TS in Table 2 is attributed to the extremely small z/L values at the end of the case, such as the night of 27–28 Dec and 29–30 Jan. Corresponding to the internal system parameters, the other common way to express the different SBL regimes is by using the external forcing parameters such as pressure gradient and cloud cover. A series of works by Van de Wiel et al. (2002a; 2002b; 2003) divided the SBL into turbulence case, intermittent case and radiative case in terms of a dimensionless number ($\Pi$) composited from the external forcing parameters. In their works the turbulent heat flux is chosen as indicator, because the turbulent heat flux is directly influenced by two external key parameters: the synoptic pressure gradient and the isothermal net radiation (Van de Wiel et al., 2003). The weakest stable case is characterized by continuously large sensible heat flux and for the intermittent case, turbulent and quiet periods alternate irregularly (see Fig. A2 in Van de Wiel et al., 2003). Figure 1 below shows the distribution of sensible heat flux at 40 m during three nights (26–27 Dec, 25–26 and 28–29 Jan). In order to maintain continuity, the early night (that is, 20:00–00:00) is also involved in Figure 1. Figure 1a and b show clear alternate occurrence of strong and weak flux, albeit Figure 1c is not so typical, suggesting intermittent sensible heat flux in the SBL.

[Figure]

Figure 1. Time series of the sensible heat flux during the nighttime of 26–27 Dec, 25–26 and 28–29 Jan.

In addition, Van de Wiel et al. (2012) and Sun et al. (2012) revealed the relationship between turbulence strength and horizontal wind speed in different stable regimes. They found that the turbulence increases slowly with increasing wind speed under a given wind speed value, and once across the threshold value, the turbulence strengthens significantly with the increase of wind speed. And this threshold value of wind speed represents the minimum wind speed needed to sustain continuous turbulence. For the case of 40 m, the minimum wind speed for the maintenance of continuous turbulence is proposed as 5 m/s. Here we briefly introduce the theory and details can refer to Van de Wiel et al. (2012). First, it is known that the high-level winds (say, > 100 m) accelerate during the nighttime while the low-level winds (say, < 20 m) weaken in the meanwhile. Hence, the wind speed at intermediate levels (say, about 30–60 m) tends to keep constant, which follows the momentum conservation principles. Mathematically, the threshold wind speed can be seen as a velocity boundary condition for the maintenance of continuous turbulence. First, we checked the distribution of 10-min averaged horizontal wind speed V at 40 m for both two cases (Figure 2). Given that all of the values of V from the pollution cases (blue solid line) are less than 5 m/s, the extra data on 22 Dec and 22 Jan are also presented for comparison (orange solid line). From Figure 2 we can see that the nighttime horizontal wind speed V during the pollution cases (blue solid line) is generally less than 5 m/s while the values of V for normal nights (orange solid line) are larger than 5 m/s. Following the analyses of Van de Wiel et al. (2012) and Sun et al. (2012), we chose the friction velocity $u_*$, the

turbulence velocity scale $V_{TKE} = \sqrt{(\sigma_u^2 + \sigma_v^2 + \sigma_w^2)/2} = \sqrt{TKE}$ (define by Sun et al., 2012), and the standard deviation of the vertical speed $\sigma_w$ to investigate the evolvement of turbulence with the increasing horizontal wind speed. Figure 3 gives the relationship between $u_*/V_{TKE}/\sigma_w$ and horizontal wind speed V during the nighttime. The relationship is similar between variables but shows two different patterns of evolvement within the wind speed range. In the case of pollution periods (blue), the horizontal wind speed is weak, accompanied by a slight increase of turbulence variables $(u_*/V_{TKE}/\sigma_w)$ with the stronger wind speed. Once the non-pollution periods (22 Dec and 22 Jan, orange) involved, turbulence strengthens rapidly with the increasing wind speed. This abrupt transform happens when the wind speed exceeds around 4 m/s which is out of the main wind-speed range that pollution samples dominate. According to Sun et al. (2012), the strong wind range (orange) in Figure 3 corresponds to their regime-2, while the weak wind part (blue) represents the regime-1 or regime-3. Whether it falls into regime-1 or regime-3 depends on the existence of upper-level motions (i.e. LLJs). As is discussed further in the last response below, LLJs play a key role in the reproduce and downward transport of turbulence in the SBL. As the wind speed is correlated with the Richardson number for very stable conditions, this wind speed threshold could serve as the transition between very stable and weakly stable conditions (Mahrt, 2014). At this point, we can summarize Figure 2–3 by concluding that the nighttime 40-m horizontal winds during both Case-1 and Case-2 are less than the minimum wind speed, so sustained and continuous turbulence is unlikely to occur in the SBL.

[Figure]

Figure 2. 10-min averaged horizontal wind speed V at 40 m for (a) Case-1 and (b) Case-2. The orange solid line represents the extra data on 22 Dec, 2016 and 22 Jan, 2017. The dashed black line marks the suggested threshold value (5 m/s).

[Figure]

Figure 3. The relationship between the friction velocity $u_*$, turbulence strength $V_{TKE}$, standard deviation of the wind speed $\sigma_w$, and horizontal wind speed V at 40 m during the nighttime. The blue plot represents samples from the pollution cases and the orange plot is based on the extra data on the night of 22 Dec, 2016 and 22 Jan, 2017. The errorbar denotes the bin-averaged results. The top panel is for Case-1 and bottom panel is for Case-2.

As Frisch (1995) stated, when all or some of the possible symmetries are restored in a statistical sense, the turbulence at high Reynolds numbers is fully-developed. The fully-developed turbulence structure is isotropic or is the same when rotated, and the probability density function (PDF) of velocity increments should be Gaussian. However, any large-scale shearing, intermittency or strongly stable stratification in the real world could prevent symmetry restoration, resulting in a distorted PDF shape from the Gaussian form. The intermittency refers to infrequent events, which can be manifested as the velocity increments furthest from zero (Vindel et al., 2008). In this case, kurtosis can serve as a useful indicator to measure the intermittency of turbulence (Vindel et al., 2008; Mahrt 2011). Since the Gaussian distribution has a Kurtosis value of 3, it can be expected that the stronger the intermittency is, the greater the tail of PDF stretches and the larger the Kurtosis values are. Although Figure S8 in the supplement has shown the distribution of kurtosis at a scale of 10 min, Figure 4 further studies the values of Kurtosis of vertical velocity increments on different scales concentrating on the three dissipation nights. The

velocity increment is defined as $u' = u_t - u_{t+i}$ ($i = 100, 200, 300, ...$), represents the fluctuation at different scales. It can be seen that for the three nights of PM dissipation, the values of Kurtosis are larger than 3 at all scales, implying that the small-scale turbulence during these periods are intermittent.

[Figure]

Figure 4. Kurtosis of vertical velocity increments at different scales. For each whisker diagram, the central rectangle spans the first quartile to the third quartile; the segment inside the rectangle shows the median and "whiskers" above and below the box show the locations of the minimum and maximum. The grey solid line denotes the Gaussian value.

The results above are added to the revision to help illustrate the intermittency of turbulence on the nights of PM dissipate and Figure 3–4 are attached in the supplement (see Figure S10–11). "*In terms of the horizontal wind speed, the 40-m weak wind less than the threshold value (i.e. 5 m s$^{-1}$) proposed by Van de Wiel et al. (2012) implies that continuous turbulence is unlikely to occur near the surface (Figure S10–11), which corresponds to the very stable turbulence regime-1 or regime-3 in Sun et al. (2012).*" (page 13, lines 6-8)

Another claim by the authors is heating at the top of the surface pollution layer by pollutants. As we know, the high pollutant concentration is highly correlated with the stable boundary layer. In other words, because of the development of the stable boundary layer, pollutants can be trapped near the surface where turbulent mixing is weak. One of the

characteristics of the stable boundary layer is air temperature increases with height. Yes, pollutants as aerosols play a role in radiative heating/cooling the atmosphere, but the authors cannot use this character of the stable boundary layer to proof the role of pollutants on radiative heating. Air temperature always increases with height in any stable boundary layer even without pollutants.

5     **Response**: Thank you for your comments. Indeed, the available results do not fully confirm the radiative effects of PM$_{2.5}$, which is also out of the scope of this work. This part has been removed from the revision.

Another claim from the authors is the role of LLJs in turbulence generation. The turbulence data used for calculating

10   their turbulence intermittency index are at 200 m and below. The lowest wind speed measurement level from the wind profiler radar is at 200 m. All the LLJs in Fig. 9 are way above 200 m. There is no way that the authors can claim turbulence is generated way above. More likely turbulence at 200 m and below is generated by wind shear between 200 m and the surface, which is supported by the increase of wind speed at about 200 m at the beginning of the TS periods in Fig. 8.

15   **Response:** Thank you for pointing out this problem. Indeed, mere height or time distribution (as in Fig. 8 and 9 in the manuscript) cannot fully address the relationship between LLJs and turbulence, not to mention that the observation of wind profile radar (WPR) below 200 m is not available. In this part, we will try to prove that: 1) the heights of LLJ 'nose' are above the top of the tower so that the limitation of vertical observation of the WPR has little effect on the detection of LLJs; 2) with the existence of LLJs, the turbulence is transported downward at

20     the levels across the tower, thus resulting in the so-called upside-down boundary layer; 3) and this kind of downward transported turbulence further confirms the intermittency of turbulence at lower levels. The relationship between LLJs and intermittent turbulence has been fully discussed in literature (Mahrt, 1979; 1999; 2014; Mahrt and Vickers, 2002; Balsley et al., 2003; Banta et al., 2006; Karipot et al., 2008). Under the influence of LLJs, the vertical structure of the SBL could be totally different from that of the traditional SBL.

25     Here we cite the schematic of these two kinds of nocturnal boundary layer from Banta et al. (2006, see their Fig.1). As we know, in a traditional SBL, the turbulence is generated by the wind shear near the surface and transported upward, as shown in the top panel of Figure 5. While in the present of LLJs, the turbulence is primarily generated at upper levels of boundary layer and then transported downward to accelerate the flows near the surface, thus changing the vertical profile of a number of turbulent variables in the SBL. For example, 1) the

30     wind shear weakens with height at lower levels and then strengthens at high levels due to the LLJ 'nose'; 2) the maximum of turbulence strength is aloft in the SBL; 3) the turbulence is transported downward and the values of vertical turbulence energy flux are negative. Based on the observation of turbulence from the tower, we can study

the vertical structure of the SBL across the tower layer. As can be seen below, although there are merely three levels observation on the tower, the basic vertical structure of the SBL is captured.

[Figure]

Figure 5. Schematic of structure of (top) traditional boundary layer vs (bottom) upside-down boundary layer by Banta et al. (2006), their Fig. 1. Left ones are mean horizontal wind speed profiles; center is the profiles of TKE or velocity-variance; and right ones represent the vertical turbulent transport of turbulence strength.

Figure 6–8 gives the distribution of wind shear, variance of vertical wind speed $\sigma_w^2$, and vertical transport of vertical velocity variance $\overline{w'^3}$ at different levels during three dissipation night. The early nights are also shown but the majority of information occurs after midnight. For all of the three nights, the wind shear at the lowest level is the strongest and decreases with height; the strength of turbulence enhances ($\sigma_w^2$) with height. In terms of vertical transport of turbulence energy, the values of $\overline{w'^3}$ across the tower layer are negative in Figure 6 and 7, implying the downward transport of turbulence. Although the variation of $\overline{w'^3}$ in Figure 8(i) is a bit of erratic, the majority of flux is downward to the lower layer.

In order to further study the general characteristics, the errorbar of variables at three levels is shown in Figure 9. From 40 m up to 120 m, the wind shear decreases, while the shear at 200 m increases again due to the influence of strong wind of the LLJ 'nose' (Figure. 9(a)–(c)). With respect to the variance of vertical wind speed $\sigma_w^2$, the

turbulence strength monotonously increases with height in Figure 9(d) and (f). The maximum at middle level in Figure 9(e) could be possibly attributed to the relative height of the subject layer and the tower height, as will be discussed further below. It is known that the wind shear at the height of LLJ 'nose' is nearly zero; meanwhile, the turbulence strength of the LLJ 'nose' is weakest due to large values of Richardson number (Banta et al., 2006). Based on the vertical distribution of wind shear and turbulence strength in Figure 9(a)–(f), it can be concluded that the height of LLJ 'nose' is well above the top of the tower. A previous study on the characteristics of LLJs in Tianjin (Wei et al., 2014) revealed that there are two major height for the occurrence of LLJs in this region: one is at 250–400 m and the other is from 1,000 to 1,300 m (see their Fig. 5(d)). At this point, we can conclude that although the observation of WPR below 200 m is not available in the present study, the detection of the LLJ 'nose' in Fig. 8 and 9 of the manuscript is uninterrupted and credible.

The primary difference between a traditional SBL and an upside-down SBL is the direction of turbulence transport. It can be seen from Figure 9(g)–(i) that the value of the vertical transport of turbulence energy $\overline{w'^3}$ are negative across the tower layer, which means the turbulence is generated at higher levels and then transported downward. Besides, although the vertical transport of turbulence energy at all three nights are downward on the average, the vertical structure varies from case to case. The magnitude of $\overline{w'^3}$ for Case-2A (Figure 9h) increases from 40 m to 120 m but then decreases again when it comes to higher layer, implying a divergence layer of the downward transport of turbulence energy between 120 m and the top of the tower, which suggests that this layer corresponds to the main source of the turbulence in the subject layer (Mahrt and Vickers, 2002). Additionally, the maximum of velocity variance $\sigma_w^2$ at 120 m in Figure 9(e) also confirms that the upper half of the tower is part of the subject layer (Banta et al., 2006). As for Case-1 in Figure 9(g), the magnitude of the vertical transport of turbulence energy monotonously decreases with height across the tower layer and so does the velocity variance $\sigma_w^2$, suggesting that the entire tower layer is below the subject layer in this case. The relative height of the LLJ and the tower is illustrated in the schematic in Figure 10. As for Case-2B, it is similar to Case-2A but not so typical. From its vertical transport of turbulence energy at 40 m (Figure 8(i)), we can see that upward and downward flux occur alternately, implying that the surface wind shear also plays a role in the enhancement of turbulence in the lower layer. But the negative mean values at three levels in Figure 9(i) indicate that the LLJ-generated turbulence dominates overall.

The LLJ-induced turbulence in the SBL further confirms the results about the turbulent intermittency in the first response. As mentioned above in Figure 3, the weak wind during the cases in the present study corresponds to the noncontinuous turbulence regime of Van de Wiel et al. (2012) or regime-1/regime-3 in Sun et al. (2012). The difference between regime-1 and regime-3 depends on whether a LLJ exists or not. In an upside-down SBL with

the present of LLJs, the turbulence intermittency can be classified as the category C turbulence intermittency brought up by Sun et al. (2012).

The mechanism of LLJ-generated intermittent turbulence is addressed in the revision and Figure 9 is added to the manuscript (see Fig. 10). "*It is well-known that LLJs are an important source of intermittent turbulence in the ABL, resulting in an 'upside-down' boundary layer structure (Mahrt, 1999; 2014; Poulos et al., 2002; Mahrt and Vickers, 2002; Banta et al., 2006; Balsley et al., 2003; Karipot et al., 2008). The vertical 'nose' shape of LLJs provides wind shear at upper levels, working as an elevated source of turbulent mixing. Then this turbulence is transported downward to the surface, resulting in non-stationary increase of turbulent mixing at lower levels. In this case, the vertical structure of 'upside-down' boundary layer is totally different from that of a traditional boundary layer. For example, (a) the wind shear decreases with height first and then increases again due to the LLJ 'nose' at upper levels; (b) the strongest turbulence is not at the surface but aloft; (c) the transport of turbulence energy is downward. Fig. 10 presents the vertical turbulence structure across the tower layer for three dissipation nights, during which LLJs occurred. From Fig. 10a–b, it can be seen that the wind shear weakens in the layer between 40 and 120 m; then it increases when it comes to higher levels. In terms of the variance of vertical wind speed $\sigma_w^2$ (Fig. 10d–f), the maximal value of turbulence strength is aloft rather than near the surface for all three nights, implying a turbulence source in mid-air. The vertical distribution of transport of turbulence energy further confirms the uplift of turbulence source. The values of the vertical transport of vertical velocity variance $\overline{w'^3}$ at three levels are negative generally, which means that the transport of turbulence energy across the tower layer is downward. It should be noticed that the magnitude of $\overline{w'^3}$ in Fig. 10h is not monotonously with height, implying a divergence layer of the downward transport of turbulence energy between 120 m and the top of the tower, which suggests that this layer corresponds to the main source of the turbulence in the subjet layer (Mahrt and Vickers, 2002). In addition, the differences in phase and strength of intermittency at three levels in Fig. 6 also confirms that the wind shear associated with the LLJ 'nose' plays an important part in the generation and transport of turbulence in the ABL.*" (page 15, lines 14-33)

[Figure]

Figure 6. Distribution of wind shear, the variance of vertical wind speed $\sigma_w^2$, the vertical transport of vertical velocity variance $\overline{w'^3}$ at three levels for Case-1.

[Figure]

Figure 7. Same as Figure 6 but for Case-2A.

[Figure]

Figure 8. Same as Figure 6 but for Case-2B.

[Figure]

Figure 9. Vertical structure of (a–c) Wind shear, (d–f) variance of vertical wind speed $\sigma_w^2$, (g–i) vertical transport of vertical velocity variance during (left) 00:00–00:06 LS on 27, Dec, 2016; (center) 00:00–00:06 LS on 26, Jan, 2017; (right) 00:00–00:06 LS on 29, Jan, 2017. The circle of errorbar denotes the mean value and the width of bar marks the standard deviation.

[Figure]

Figure 10. Schematic of the relative location of LLJ and turbulence for the night of (a) Case-1 and (b) Case-2A.

**Change 2.** The statement about the radiative effect of pollutants is removed from the revision.

**Change 3.** The vertical structure of the SBL within the tower height based on the three-level turbulence observation reveals that 1) the absent of observation below 200m of the WPR does not affect the detection of LLJ 'nose'; 2) the turbulence is generated aloft and transported downward to the surface during the periods of dissipation due to the existence of LLJs; 3) the downward transport of turbulence verifies that the turbulence in the SBL is intermittent essentially.

**Marked-up manuscript version**

[revised manuscript text omitted]

 $179 \pm 85$ | $104 \pm 138$
 $75 \pm 92$ |
| Temperature (K) | $275.6 \pm 2.9$ | $279.2 \pm 0.9$ | $272.7 \pm 2.3$
 $275.6 \pm 0.8$ | $275.4 \pm 1.8$
 $273.1 \pm 1.4$ |
| RH (%) | $53 \pm 15$ | $32 \pm 8$ | $54 \pm 12$
 $48 \pm 15$ | $35 \pm 17$
 $22 \pm 2$ |
| U (m s$^{-1}$) | $1.85 \pm 1.38$ | $3.78 \pm 2.90$ | $0.61 \pm 0.87$
 $0.67 \pm 2.03$ | $1.47 \pm 1.39$
 $3.23 \pm 1.98$ |
| Magnitude of W (m s$^{-1}$) | $0.28 \pm 0.26$ | $0.61 \pm 0.64$ | $0.23 \pm 0.22$
 $0.29 \pm 0.29$ | $0.36 \pm 0.35$
 $0.62 \pm 0.58$ |
| $u_*$ (m s$^{-1}$) | $0.25 \pm 0.12$ | $0.59 \pm 0.26$ | $0.19 \pm 0.10$
 $0.29 \pm 0.13$ | $0.33 \pm 0.15$
 $0.58 \pm 0.18$ |
| TKE (m$^2$ s$^{-2}$) | $0.50 \pm 0.43$ | $3.35 \pm 2.81$ | $0.28 \pm 0.93$
 $0.25 \pm 0.71$ | $0.54 \pm 0.42$

[revised manuscript text omitted]

---

## Author Response (AR3)

**Response**

Dear Editor,

The manuscript has been revised according to the submission guidelines on the ACP website and proof-read the manuscript to minimize typographical, grammatical and bibliographical errors.

**List of relevant changes in the manuscript**

**Change 1.** Figure 1 has been replotted to avoid the parallel usage of green and red.
**Change 2.** We have revised some format errors, including units, figures, and abbreviations and so on.

**Marked-up manuscript version**

[revised manuscript text omitted]

| PM2.5 ($\mu$g m$^{-3}$) | 145 ± 71 | 31 ± 35 | 139 ± 101
± 85 | 104 ± 138
± 92 |
| Temperature (K) | 275.6 ± 2.9 | 279.2 ± 0.9 | 272.7 ± 2.3
275.6 ± 0.8 | 275.4 ± 1.8
273.1 ± 1.4 |
| RH (%) | 53 ± 15 | 32 ± 8 | 54 ± 12
± 15 | 35 ± 17
± 2 |
| U (m s$^{-1}$) | 1.85 ± 1.38 | 3.78 ± 2.90 | 0.61 ± 0.87
0.67 ± 2.03 | 1.47 ± 1.39
3.23 ± 1.98 |
| Magnitude of W (m s$^{-1}$) | 0.28 ± 0.26 | 0.61 ± 0.64 | 0.23 ± 0.22
0.29 ± 0.29 | 0.36± 0.35
0.62 ± 0.58 |
| $u_*$ (m s$^{-1}$) | 0.25 ± 0.12 | 0.59 ± 0.26 | 0.19 ± 0.10 | 0.33 ± 0.15 |

[revised manuscript text omitted]